# Antiseizure medication use during pregnancy and children's neurodevelopmental outcomes

Paul Madley-Dowd [1,2,3,17] ✉, Viktor H. Ahlqvist [4,5,17] ✉, Harriet Forbes[1,6], Jessica E. Rast[7,8], Florence Z. Martin [1,2], Caichen Zhong[7], Ciarrah-Jane S. Barry [1,2], Daniel Berglind[9,10], Michael Lundberg[9], Kristen Lyall[8], Craig J. Newschaffer[11], Torbjörn Tomson[12], Neil M. Davies [13,14,15], Cecilia Magnusson[9,10,18], Dheeraj Rai [1,2,3,16,18] & Brian K. Lee[7,8,9,18]

The teratogenic potential of valproate in pregnancy is well established; however, evidence regarding the long-term safety of other antiseizure medications (ASMs) during pregnancy remains limited. Using routinely collected primary care data from the UK and nationwide Swedish registries to create a cohort of 3,182,773 children, of which 17,495 were exposed to ASMs in pregnancy, we show that those exposed to valproate were more likely to receive a diagnosis of autism, intellectual disability, and ADHD, when compared to children not exposed to ASMs. Additionally, children exposed to topiramate were 2.5 times more likely to be diagnosed with intellectual disability (95% CI: 1.23–4.98), and those exposed to carbamazepine were 1.25 times more likely to be diagnosed with autism (95% CI: 1.05–1.48) and 1.30 times more likely to be diagnosed with intellectual disability (95% CI: 1.01–1.69). There was little evidence that children exposed to lamotrigine in pregnancy were more likely to receive neurodevelopmental diagnoses. While further research is needed, these findings may support considering safer treatment alternatives well before conception when clinically appropriate.

Antiseizure medications (ASMs) play an important role in managing seizures in epilepsy and mood stabilization in psychiatric conditions. However, using an ASM during pregnancy presents complex challenges due to risks to fetal development. Some ASMs, such as valproate, are well-established teratogens and contraindicated in pregnancy regardless of the indication[1,2].

Mounting evidence suggests that certain ASMs are associated with both fetal malformations and the long-term neurodevelopment of

[1]Centre for Academic Mental Health, Population Health Sciences, Bristol Medical School, University of Bristol, Bristol, UK. [2]Medical Research Council Integrative Epidemiology Unit at the University of Bristol, Bristol, UK. [3]NIHR Biomedical Research Centre, University of Bristol, Bristol, UK. [4]Institute of Environmental Medicine, Karolinska Institutet, Stockholm, Sweden. [5]Department of Biomedicine, Aarhus University, Aarhus, Denmark. [6]Faculty of Epidemiology and Population Health, London School of Hygiene and Tropical Medicine, London, UK. [7]A.J. Drexel Autism Institute, Drexel University, Philadelphia, PA, USA. [8]Department of Epidemiology and Biostatistics, Drexel University Dornsife School of Public Health, Philadelphia, PA, USA. [9]Department of Global Public Health, Karolinska Institutet, Stockholm, Sweden. [10]Centre for Epidemiology and Community Medicine, Region Stockholm, Stockholm, Sweden. [11]College of Health and Human Development, The Pennsylvania State University, Philadelphia, PA, USA. [12]Department of Clinical Neuroscience, Karolinska Institute, Stockholm, Sweden. [13]K.G. Jebsen Center for Genetic Epidemiology, Department of Public Health and Nursing, Norwegian University of Science and Technology, Trondheim, Norway. [14]Division of Psychiatry, University College London, London, UK. [15]Department of Statistical Science, University College London, London, UK. [16]Avon and Wiltshire Partnership NHS Mental Health Trust, Bristol, UK. [17]These authors contributed equally: Paul Madley-Dowd, Viktor H. Ahlqvist. [18]These authors jointly supervised this work: Cecilia Magnusson, Dheeraj Rai, Brian K. Lee. ✉e-mail: p.madley-dowd@bristol.ac.uk; viktor.ahlqvist@ki.se

offspring[3–10]. Notably, landmark studies such as the Neurodevelopmental Effects of Antiepileptic Drugs study (N = 311) have demonstrated that children exposed to valproate in utero have lower IQs at age six[5]. However, less is known about the long-term effects of newer or less commonly used ASMs[11,12].

Recent research[13–17] (see Supplementary Background), including a large-scale Nordic study[15], has raised concerns about the potential risks of drugs other than valproate. Amongst several findings, the Nordic study[15] reported a twofold increase in the rate of neurodevelopmental conditions among children born to mothers with epilepsy who were prescribed topiramate during pregnancy versus no ASM (HR 2.13, 95% CI 1.13–4.01). This finding could not be replicated by a large-scale US study of autism[18], where the association between topiramate and children's autism diminished after controlling for confounders. However, the small number of pregnancies exposed to monotherapy topiramate (N = 246 in the Nordic[15] & N = 623 in the US[18]), the few cases of neurodevelopmental outcomes (N = 10 in the Nordic[15] & N < 11 in the US[18]) and the short follow-up of the children (median 5.7 years in the Nordic[15] & 2 years in the US study[18]), mean more evidence is needed from large populations with longitudinal follow-up about the risks associated with these medications.

Determining whether these associations are due to causal effects of the medications, or confounding is extremely challenging because randomized controlled trials have not been possible for practical and ethical reasons[11,12]. Confounding is especially concerning since the indications for ASMs are likely to be affected by similar genetic variation to neurodevelopmental conditions[19]. Despite considerable research attention, few studies have been able to credibly identify the causal effects of these medications distinct from genetic confounding.

To contribute to the current evidence and aid ongoing reviews by regulatory agencies, such as the European Medicines Agency[20] and the Medicines and Healthcare Products Regulatory Agency (MHRA)[21], we conducted a study combining evidence from the UK primary care-based Clinical Practice Research Datalink (CPRD) GOLD and nationwide Swedish registries. By meta-analyzing across these datasets, we constructed a large-scale cohort of exposed children and accounted for observed confounding through conventional epidemiological methods, and time-invariant unobserved confounding by comparing exposed and unexposed siblings.

## Results
### Descriptives
Among the 3,182,773 children, 17,495 were exposed to ASMs (Table 1). ASM users had more health care visits, co-prescription of antidepressants and antipsychotics, were more likely to have a neurodevelopmental diagnosis, and were more likely to have given birth in more recent years. There were 78,442 children diagnosed with autism, 26,787 diagnosed with intellectual disability, and 155,329 diagnosed with attention deficit hyperactivity disorder (ADHD). Table S1 shows the number exposed to each ASM and summary statistics for the length of follow-up in each country.

### Primary analyzes: ASM use and offspring neurodevelopmental conditions
After adjusting for differences in covariates between children unexposed and exposed to ASMs during pregnancy, and meta-analyzing across countries, at age 12 the expected absolute risk of autism, intellectual disability, and ADHD among children unexposed to ASMs was 3.01% (95% CI: 2.94–3.08), 0.92% (95% CI: 0.90–0.94), and 2.50% (95% CI: 2.39–2.63), respectively (Fig. 1). The ASM with the highest expected absolute risk varied by outcome. For autism, the highest absolute risk at age 12 was observed for valproate monotherapy (4.85%; 95% CI: 4.07–5.78), followed by polytherapy (4.29%; 95% CI: 3.49–5.28) and gabapentin monotherapy (3.67%: 95% CI: 2.70–4.09). For intellectual disability, the greatest risk at age 12 was observed for

valproate monotherapy (2.36%; 95% CI: 1.83–3.05), topiramate monotherapy (2.14%: 95% CI: 1.08–4.24) and polytherapy (1.80%; 95% CI: 1.33–2.43). For ADHD the greatest risk at age 12 was observed for valproate (6.34%; 95% CI: 5.32–7.56), phenytoin (6.30%: 95% CI: 4.38–9.07), and polytherapy (5.88%: 95% CI: 4.78–7.22). Combined and country-specific values for all ASMs, as well as risk difference values are presented in Supplementary Results Tables S2–5 and Figs. S3–6.

Figure 2 shows meta-analyzed hazard ratios (HR) of each outcome for each ASM monotherapy compared to no ASM exposure during pregnancy. Children exposed to valproate (HR: 1.78; 95% CI: 1.48–2.14), polytherapy (HR: 1.51; 95% CI: 1.22–1.88), and carbamazepine (HR: 1.25; 95% CI: 1.05–1.48) were more likely to have an autism diagnosis than unexposed children. Children exposed to these drugs also had an increased hazard of intellectual disability (valproate HR: 2.56, 95% CI:1.97–3.32; polytherapy HR: 2.02, 95% CI: 1.49–2.75; carbamazepine HR: 1.30, 95% CI: 1.01–1.69) in addition to increased hazard associated with topiramate (HR: 2.48; 95% CI: 1.23–4.98). Children exposed to valproate (HR: 1.20; 95% CI: 1.02–1.40) had an increased hazard of ADHD, while children exposed to pregabalin had decreased hazard of ADHD (HR: 0.67; 95% CI: 0.55–0.83). Country-specific and combined HRs for all ASM monotherapies are presented in Supplementary Results Table S6 and Fig. S7. There was little evidence to suggest that the estimates in primary analyzes differed between countries, except for the HR of ADHD for lamotrigine where the estimate in the UK was greater than in Sweden (see Supplementary Results Table S7).

### Secondary analyzes: sibling, active comparator and indication restricted analyzes
Sibling analyzes, presented in Fig. 3A, account for unmeasured genetic and environmental confounding shared between siblings. Results were largely consistent with the primary analyzes (see Supplementary Results Table S8), except for phenytoin where large HRs were found for autism (within-family HR: 11.53; 95% CI: 4.00–33.20) and ADHD (within-family HR: 9.48; 95% CI: 3.60–24.95). These estimates were heavily weighted towards CPRD, where there were a small number of exposed individuals (see Table S1). Within-family HRs for ADHD were also not consistent for pregabalin, other ASMs and polytherapy. Country-specific and combined values are presented in Supplementary Results Table S8 and Fig. S8.

We present our active comparison analysis, using lamotrigine as the reference group, in Fig. 3B. This analysis makes a comparison with children exposed to an ASM considered to be safe, thereby reducing the possibility of confounding by indication. Our analyzes showed that, relative to lamotrigine, children exposed to carbamazepine, valproate, other ASMs, and polytherapy had greater hazard for a diagnosis of autism. Children exposed to topiramate, valproate and polytherapy had greater hazard of an intellectual disability diagnosis than those exposed to lamotrigine. Children exposed to valproate and polytherapy had greater hazard of an ADHD diagnosis, while children exposed to pregabalin were less likely to receive an ADHD diagnosis. Country-specific and combined values are presented in Supplementary Results Table S9 and Fig. S9.

Indication restricted analyzes (including drug specific estimates) are presented in Table 2, and Supplementary Results Tables S10–S12 and Figs. S10–S14. Children born to mothers with epilepsy were more likely to be diagnosed with autism, ID, and ADHD at age 12, irrespective of ASM exposure (unexposed risk: autism 5.03%; 95% CI: 4.45–5.69, intellectual disability 1.91%; 95% CI: 1.62–2.25, ADHD 10.77%; 95% CI: 9.85–11.77), when compared to the risk in the entire population. Similarly, children born to mothers with psychiatric indications (unexposed risk: autism 6.11%; 95% CI: 5.82–6.42, intellectual disability 1.19%; 95% CI: 1.11–1.29, ADHD 3.87%; 95% CI: 3.60–4.17) and children born to mothers with somatic indications (unexposed risk: autism 5.54%; 95% CI: 5.15–5.96, intellectual disability 1.15%; 95% CI: 1.05–1.25, ADHD 3.87%; 95% CI: 3.44–4.36) exhibited a higher risk, irrespective ASM exposure,

**Table 1 | Characteristics of offspring included at each data source, by ASM exposure during pregnancy**

| | UK - CPRD | | Sweden - DOHaD | |
|---|---|---|---|---|
| | Unexposed to ASM (*N* = 514,066) | Exposed to ASM (*N* = 3981) | Unexposed to ASM (*N* = 2,651,212) | Exposed to ASM (*N* = 13,514) |
| **Offspring female sex at birth**, N (%) | 250,213 (48.7) | 1915 (48.1) | 1,287,812 (48.6) | 6850 (50.7) |
| **Parity**, N (%) | | | | |
| 1 | 300,841 (58.5) | 2287 (57.4) | 1,136,824 (42.9) | 6204 (45.9) |
| 2 | 138,190 (26.9) | 1014 (25.5) | 975,646 (36.8) | 4347 (32.2) |
| 3 | 49,684 (9.7) | 421 (10.6) | 371,335 (14.0) | 1945 (14.4) |
| 4 | 16,814 (3.3) | 159 (4.0) | 106,943 (4.0) | 623 (4.6) |
| 5+ | 8537 (1.7) | 100 (2.5) | 60,464 (2.3) | 395 (2.9) |
| **Maternal ASM indication pre-pregnancy[a] (not mutually exclusive groups)**, N (%) | | | | |
| Epilepsy | 4075 (0.8) | 2433 (61.1) | 10,769 (0.4) | 7016 (51.9) |
| Psychiatric indications | 183,004 (35.6) | 2465 (61.9) | 189,904 (7.2) | 558 (41.3) |
| Somatic indications | 69,091 (13.4) | 1188 (29.8) | 244,909 (9.2) | 3435 (25.4) |
| No identified indication | 297,528 (57.9) | 137 (3.4) | 2,253,836 (85.0) | 1489 (11.0) |
| **Maternal age at delivery, mean (SD)** | 29.1 (5.8) | 29.6 (5.8) | 30.6 (5.1) | 30.8 (5.3) |
| **Maternal cohabitation at delivery**, N (%) | N/A | N/A | 2,386,438 (90.0) | 11,042 (81.7) |
| **Maternal country of birth is Sweden**, N (%) | N/A | N/A | 2,054,232 (77.5) | 11,494 (85.1) |
| **Residential region at pregnancy start[b]**, N (%) | | | | |
| 1 | 352,909 (68.7) | 2297 (57.7) | 224,611 (8.5) | 1274 (9.4) |
| 2 | 26,925 (5.2) | 321 (8.1) | 517,025 (19.5) | 3166 (23.4) |
| 3 | 83,244 (16.2) | 839 (21.1) | 667,848 (25.2) | 2732 (20.2) |
| 4 | 50,988 (9.9) | 524 (13.2) | 270,534 (10.2) | 1531 (11.3) |
| 5 | – | – | 533,771 (20.1) | 2762 (20.4) |
| 6 | – | – | 437,423 (16.5) | 2049 (15.2) |
| **Index of Multiple Deprivation (IMD) / Household disposable income[c], quintiles** N (%) | | | | |
| 1 – Least deprived/Highest income | 99,132 (19.3) | 587 (14.7) | 572,017 (21.6) | 2119 (15.7) |
| 2 | 92,169 (17.9) | 608 (15.3) | 55,547 (21.0) | 239 (17.7) |
| 3 | 99,572 (19.4) | 726 (18.2) | 54,345 (20.5) | 2923 (21.6) |
| 4 | 101,363 (19.7) | 892 (22.4) | 519,354 (19.6) | 349 (25.8) |
| 5 – Most deprived/Lowest income | 121,830 (23.7) | 1168 (29.3) | 460,921 (17.4) | 2592 (19.2) |
| **Highest household education**, N (%) | | | | |
| Secondary schooling level (mandatory) | N/A | N/A | 139,191 (5.3) | 1124 (8.3) |
| Upper secondary level | N/A | N/A | 1,040,510 (39.2) | 6398 (47.3) |
| University level | N/A | N/A | 1,471,511 (55.5) | 5992 (44.3) |
| **History of addiction before pregnancy**, N (%) | 5723 (1.1) | 109 (2.7) | 61,295 (2.3) | 2092 (15.5) |
| **Any maternal neurodevelopmental conditions diagnosed before birth of offspring** *N* (%) | 1359 (0.26) | 56 (1.41) | 74,372 (2.8) | 2368 (17.5) |
| **Health care visits in year before pregnancy**, N (%) | | | | |
| 0–3 | 175,360 (34.1) | 364 (9.1) | 2,442,450 (92.1) | 8862 (65.6) |
| 4–10 | 223,816 (43.5) | 1268 (31.9) | 185,939 (7.0) | 3684 (27.3) |
| 11+ | 114,890 (22.3) | 2349 (59.0) | 22,823 (0.9) | 968 (7.2) |
| **Seizure recorded in the year before pregnancy**, N (%) | 264 (0.1) | 289 (7.3) | 1545 (0.1) | 2471 (18.3) |
| **Antipsychotic use[d] in the year before pregnancy**, N (%) | 320 (0.1) | 21 (0.5) | 29,623 (1.1) | 1921 (14.2) |
| **Antidepressant use[d] in the year before pregnancy**, N (%) | 47,214 (9.2) | 1251 (31.4) | 87,593 (3.3) | 3464 (25.6) |
| **Antiemetics use during pregnancy**, N (%) | 64,045 (12.5) | 903 (22.7) | 1132 (0.4) | 212 (1.6) |
| **Calendar year of delivery**, N (%) | | | | |
| 1995–97 | 10,682 (2.1) | 64 (1.6) | 262,871 (9.9) | 475 (3.5) |
| 1998–00 | 29,148 (5.7) | 153 (3.8) | 253,599 (9.6) | 682 (5.0) |
| 2001–03 | 54,801 (10.7) | 280 (7.0) | 274,811 (10.4) | 834 (6.2) |
| 2004–06 | 96,758 (18.8) | 593 (14.9) | 297,701 (11.2) | 979 (7.2) |
| 2007–09 | 102,497 (19.9) | 679 (17.1) | 314,565 (11.9) | 1488 (11.0) |
| 2010–12 | 98,664 (19.2) | 727 (18.3) | 328,361 (12.4) | 1989 (14.7) |
| 2013–15 | 73,119 (14.2) | 780 (19.6) | 332,953 (12.6) | 2274 (16.8) |
| 2016–18 | 48,397 (9.4) | 705 (17.7) | 340,461 (12.8) | 2766 (20.5) |
| 2019–20 | – | – | 245,890 (9.3) | 2027 (15.0) |

*ASM* antiseizure medication, *CPRD* Clinical Practice Research Datalink, *DOHaD* Developmental Origins of Health and Disease, *SD* standard deviation.

[a]In the UK, somatic indications: migraine, neuropathic pain (including diabetic neuropathy) and fibromyalgia, restless legs syndrome, essential tremors, and psychiatric indications: bipolar disorder, generalized anxiety disorder, severe depression, treatment-resistant schizophrenia, other off-label psychiatric presumed indications; In Sweden, somatic indications: migraine, chronic pain (including, for example, fibromyalgia), diabetes and related neuropathy, and psychiatric indications: bipolar disorder, depression, generalized anxiety disorder, and any other psychiatric conditions.

[b]Regions in the UK: 1 = England, 2 = Northern Ireland, 3 = Scotland, and 4 = Wales; regions in Sweden: 1 = Northern Sweden, 2=Middle Sweden, 3 = Stockholm, 4 = Southeast Sweden, 5 = West Sweden, 6 = Southern Sweden.

[c]In the UK, we use the IMD, while in Sweden, we use household disposable income.

[d]Prescribed in the UK and either dispensed or from antenatal visits in Sweden.

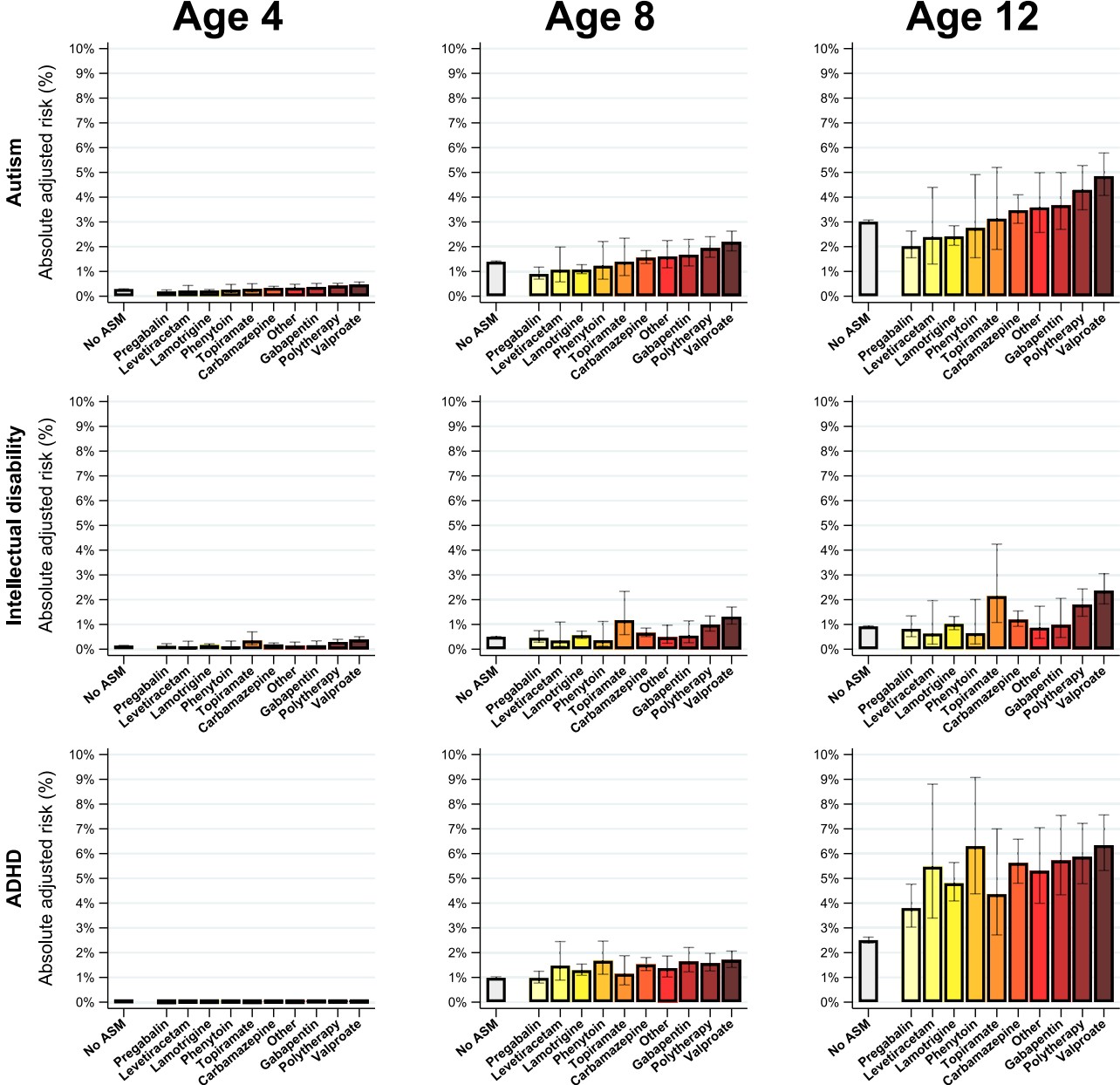

**Fig. 1 | The pooled absolute adjusted risk (%) of offspring neurodevelopmental conditions by ASM exposure during pregnancy at ages 4, 8, and 12.** All estimates are adjusted for maternal age, region, diagnosis of neurodevelopmental conditions before pregnancy, evidence of hazardous drinking and illicit drug use during pregnancy, gravidity, health care utilization, seizure events, use of antipsychotics and antidepressants in the year prior to pregnancy, vomiting or antiemetic prescriptions during pregnancy, and socioeconomic position. Data are presented as pooled absolute adjusted risk (end line of bars) ± 95% confidence limits (grey error bars). Presented measures were estimated using fixed-effects meta-analysis on the log-risk scale. Note that pooled estimates of the risk of ADHD among those not exposed to an ASM are heavily weighted towards the UK (risk age 12 = 2.10; 95% CI = 1.99–2.21), where a lower risk was estimated than in Sweden (risk age 12 = 5.70; 95% CI = 5.09–6.38), due to the calculation of lower standard errors for estimated risks closer to 0 (see Supplementary Results Tables S2 and S3). Sample size for figure: Total cohort = 3,182,771, No ASM = 3,165,276, Carbamazepine = 3030, Gabapentin = 1428, Lamotrigine = 5974, Levetiracetam = 806, Phenytoin = 240, Pregabalin = 1715, Topiramate = 418, Valproate = 1601, Other ASM = 543, Polytherapy = 1740. Source data are provided as a Source Data file. ADHD attention deficit hyperactivity disorder, ASM antiseizure medication.

when compared to the general population. We recommend caution when interpreting results from indication restricted analysis as we have shown that, in the presence of unmeasured confounding, these may be more biased than standard analysis[22].

**Sensitivity analyzes**
Results of sensitivity analyzes 1–3 are presented in Supplementary Results Table S13. Briefly, the findings were largely consistent with the

primary analysis, albeit some values were estimated with more uncertainty. That is, restricting the cohorts to ensure a minimum follow-up of 4 years did not change the conclusions of our primary analyzes. When exploring exposure during the first trimester, all ASM and NDD combinations were comparable to the primary analyzes except for phenytoin, which became associated with ADHD with large uncertainty. When modifying the exposure definition to require two prescriptions/dispensations, the results were comparable with the

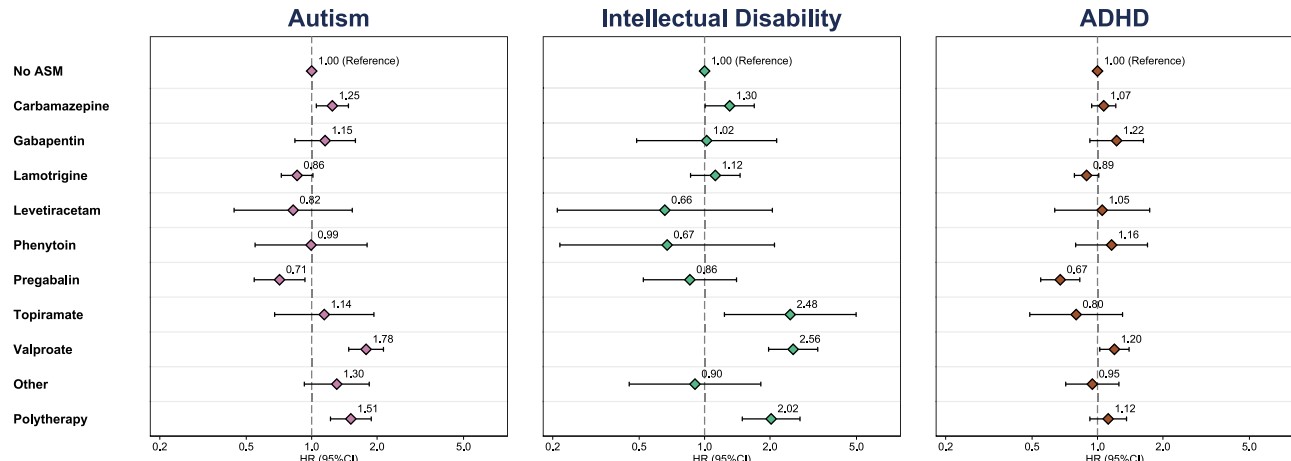

**Fig. 2 | The pooled adjusted hazard ratios of offspring neurodevelopmental conditions by ASM exposure during pregnancy.** All estimates are adjusted for maternal age, region, diagnosis of neurodevelopmental conditions before pregnancy, evidence of hazardous drinking and illicit drug use during pregnancy, gravidity, health care utilization, seizure events, use of antipsychotics and antidepressants in the year prior to pregnancy, vomiting or antiemetic prescriptions during pregnancy, and socioeconomic position. Error bars show 95% confidence limits. The presented measures were estimated using fixed-effects meta-analysis on the log-hazard ratio scale. Sample size for figure: Total cohort = 3,182,771, No ASM = 3,165,276, Carbamazepine = 3030, Gabapentin = 1428, Lamotrigine = 5974, Levetiracetam = 806, Phenytoin = 240, Pregabalin = 1715, Topiramate = 418, Valproate = 1601, Other ASM = 543, Polytherapy = 1740. Source data are provided as a Source Data file. ADHD attention deficit hyperactivity disorder, ASM antiseizure medication, HR hazard ratio.

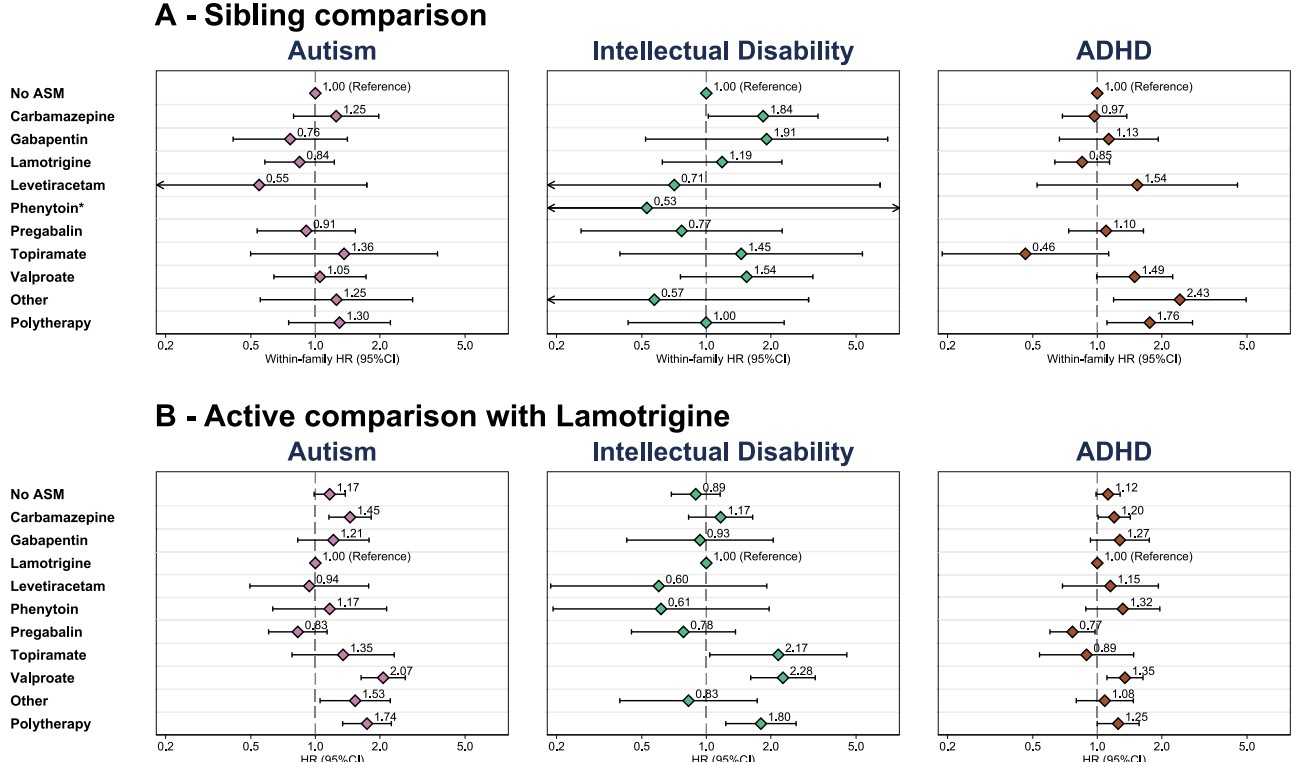

**Fig. 3 | Sibling comparisons and active comparison with lamotrigine by ASM exposure during pregnancy.** The pooled estimates for (**A**) sibling comparisons (within-family hazard ratio) and (**B**) active comparison with lamotrigine (hazard ratio) by ASM prescription during pregnancy. All estimates are adjusted for maternal age, region, diagnosis of neurodevelopmental conditions before pregnancy, evidence of hazardous drinking and illicit drug use during pregnancy, gravidity, health care utilization, seizure events, use of antipsychotics and antidepressants in the year prior to pregnancy, vomiting or antiemetic prescriptions during pregnancy, and socioeconomic position. Error bars show 95% confidence limits. The presented measures were estimated using fixed-effects meta-analysis on the log-hazard ratio scale. Sample size for figure: Total cohort = 3,182,771, No ASM = 3,165,276, Carbamazepine = 3030, Gabapentin = 1428, Lamotrigine = 5974, Levetiracetam = 806, Phenytoin = 240, Pregabalin = 1715, Topiramate = 418, Valproate = 1601, Other ASM = 543, Polytherapy = 1740. * Sibling comparison results for phenytoin are presented in the text. Source data are provided as a Source Data file. ADHD attention deficit hyperactivity disorder, ASM antiseizure medication, HR hazard ratio.

**Table 2 | The pooled adjusted hazard ratios of offspring neurodevelopmental conditions by ASM exposure during pregnancy, stratified according to maternal diagnosis of epilepsy, and psychiatric or somatic indications**

| | Epilepsy | | Psychiatric indication | | Somatic indication | |
|---|---|---|---|---|---|---|
| | Weight for Sweden (%) | Pooled hazard ratio (95% CI) | Weight for Sweden (%) | Pooled hazard ratio (95% CI) | Weight for Sweden (%) | Pooled hazard ratio (95% CI) |
| **Autism** | | | | | | |
| No ASM | – | 1.00 (Reference) | – | 1.00 (Reference) | – | 1.00 (Reference) |
| Carbamazepine | 87.8 | 1.21 (0.98–1.50) | 72.7 | 1.37 (0.99–1.90) | 86.7 | 1.46 (0.98–2.19) |
| Gabapentin | 61 | 0.99 (0.21–4.60) | 31.1 | 1.24 (0.82–1.88) | 43.7 | 1.27 (0.83–1.96) |
| Lamotrigine | 77.5 | 0.86 (0.67–1.12) | 88.9 | 1.02 (0.83–1.26) | 86 | 0.88 (0.62–1.23) |
| Levetiracetam | 82.7 | 0.94 (0.49–1.79) | 100 | 1.18 (0.38–3.72) | ** | ** |
| Phenytoin | 87.2 | 0.84 (0.42–1.71) | 0[a] | 2.96 (0.46–19.27) | 100 | 1.18 (0.29–4.86) |
| Pregabalin | ** | ** | 84.8 | 0.84 (0.61–1.15) | 83.3 | 0.80 (0.54–1.21) |
| Topiramate | 86.6 | 2.30 (1.08–4.92) | 100 | 0.96 (0.43–2.15) | 100 | 1.03 (0.38–2.74) |
| Valproate | 87.5 | 2.04 (1.62–2.55) | 72.2 | 1.57 (1.13–2.19) | 69 | 1.55 (0.98–2.47) |
| Other | 100 | 1.53 (0.89–2.61) | 88.7 | 1.44 (0.89–2.33) | 90.2 | 1.57 (0.83–2.94) |
| Polytherapy | 79.1 | 1.90 (1.49–2.43) | 67.9 | 1.59 (1.16–2.18) | 71.9 | 1.21 (0.78–1.87) |
| **Intellectual disability** | | | | | | |
| No ASM | – | 1.00 (Reference) | | 1.00 (Reference) | – | 1.00 (Reference) |
| Carbamazepine | 94.9 | 1.46 (1.08–1.97) | 86.9 | 0.75 (0.37–1.54) | 100 | 1.88 (1.03–3.42) |
| Gabapentin | ** | ** | 100 | 1.04 (0.33–3.23) | 100 | 1.09 (0.41–2.91) |
| Lamotrigine | 90.1 | 1.26 (0.88–1.81) | 93.9 | 1.24 (0.87–1.77) | 87.4 | 1.58 (0.97–2.58) |
| Levetiracetam | 100 | 0.90 (0.28–2.83) | ** | ** | ** | ** |
| Phenytoin | 100 | 0.73 (0.23–2.30) | ** | ** | 100 | 1.36 (0.19–10.04) |
| Pregabalin | ** | ** | 100 | 1.04 (0.60–1.81) | 100 | 0.87 (0.41–1.84) |
| Topiramate | 100 | 5.73 (2.31–14.21) | 67.1 | 1.74 (0.56–5.45) | 50.8 | 1.99 (0.49–8.06) |
| Valproate | 96.2 | 2.73 (2.00–3.72) | 81.8 | 1.56 (0.85–2.85) | 100 | 1.64 (0.71–3.75) |
| Other | 100 | 1.08 (0.44–2.64) | 100 | 0.55 (0.14–2.21) | 100 | 0.86 (0.21–3.47) |
| Polytherapy | 89.5 | 2.65 (1.89–3.70) | 70.4 | 1.29 (0.71–2.36) | 75.8 | 2.00 (1.09–3.69) |
| **ADHD** | | | | | | |
| No ASM | – | 1.00 (Reference) | – | 1.00 (Reference) | – | 1.00 (Reference) |
| Carbamazepine | 94.9 | 1.04 (0.89–1.22) | 76.7 | 1.05 (0.79–1.40) | 94.2 | 0.96 (0.68–1.37) |
| Gabapentin | 100 | 1.62 (0.67–3.93) | 57.7 | 1.26 (0.82–1.93) | 76.1 | 1.59 (1.11–2.30) |
| Lamotrigine | 91.5 | 0.87 (0.71–1.06) | 95.9 | 1.01 (0.85–1.20) | 94.3 | 0.84 (0.63–1.13) |
| Levetiracetam | 93.6 | 1.00 (0.60–1.68) | 69.4 | 1.31 (0.58–3.00) | 31.7 | 3.34 (1.53–7.32) |
| Phenytoin | 96.1 | 1.18 (0.78–1.78) | 67.8 | 1.61 (0.51–5.11) | 100 | 0.80 (0.25–2.53) |
| Pregabalin | 0[a] | 35.67 (8.47–150.20) | 96.7 | 0.73 (0.56–0.94) | 97.8 | 0.90 (0.66–1.21) |
| Topiramate | 100 | 1.01 (0.38–2.71) | 89.5 | 0.78 (0.41–1.51) | 88.2 | 0.94 (0.47–1.89) |
| Valproate | 95.2 | 1.22 (1.01–1.48) | 89.8 | 1.06 (0.79–1.44) | 88.7 | 0.67 (0.41–1.11) |
| Other | 100 | 1.11 (0.74–1.67) | 95.8 | 1.07 (0.71–1.62) | 100 | 1.21 (0.72–2.02) |
| Polytherapy | 93.7 | 1.15 (0.91–1.45) | 91.9 | 1.18 (0.88–1.60) | 93.2 | 0.98 (0.66–1.47) |

**Could not be estimated due to no observed events.
[a] There were no observed cases in Sweden, and <5 in the UK.

primary analyzes for all ASM-outcome combinations, except topiramate and intellectual disability where the estimate was increased under the new definition. Repeating primary analyzes excluding the covariate for vomiting or antiemetic prescriptions from the adjustment set only changed hazard ratio estimates beyond the second decimal (see Supplementary Results Table S14).

## Discussion

The findings of this large-scale study support prior findings that valproate, topiramate, carbamazepine, and polytherapy use during pregnancy are associated with neurodevelopmental conditions in offspring. For example, children exposed to topiramate during pregnancy were 2.5 times more likely to be diagnosed with intellectual disability, as compared to children not exposed to ASMs, corresponding to an absolute risk increase of 1.2% at age 12. Our findings

also found little evidence that *in utero* exposure to lamotrigine increase the risk of neurodevelopmental conditions.

**Comparison with previous literature**
Our study represents one of the largest studies of children diagnosed with neurodevelopmental conditions after ASM exposure in pregnancy, as most previous studies have been based on few exposed children diagnosed with neurodevelopmental conditions[13–18]. We replicated the association between *in utero* exposure to topiramate and intellectual disability found in a recent Nordic study[15], but did not replicate an association with autism in the general population. Our findings are consistent with those of a recent US study of autism diagnoses in children exposed to toprimate[18]. In contrast to recent findings, we found little evidence that children exposed to topiramate or levetiracetam were more likely to be diagnosed with ADHD in the

general population or in children of mothers with epilepsy[16]. Previous findings for carbamazepine have been conflicting; associations have been found with autism[15] and intellectual disability[14,15] (though only when not accounting for indication), but not in all studies and not for ADHD[3,16]. In the Supplementary Background, we briefly summarize recent studies on carbamazepine and neurodevelopmental conditions (Supplementary Background Fig. 1); the association across recent studies appears largely consistent with ours, albeit some studies have not reported statistically significant associations, possibly because the number of exposed cases have been smaller than presented here. Our study provides evidence suggesting that carbamazepine is associated with an increased likelihood of autism and intellectual disability after accounting for indication through multivariable regression, albeit there is still some uncertainty in the indication stratified data.

Specific polytherapy combinations, including levetiracetam with carbamazepine and lamotrigine with topiramate, have been suggested to be associated with neurodevelopmental conditions. Our results suggest an increased risk of autism and intellectual disability in the children of mothers exposed to polytherapy during pregnancy, though we did not explore specific combinations. As valproate and carbamazepine were more commonly used in the polytherapy-exposed group, this may account for the observed excess risks (see Supplementary Results Table S15). It is worth caveating that such findings may be particularly susceptible to confounding by severity of indication as mothers taking more than one ASM may be more likely to suffer from more severe epilepsy or other indications than mothers taking monotherapy.

Valproate has been consistently associated with neurodevelopmental conditions in offspring[3,7,9,10,14–16,18]. Our findings support this for intellectual disability, ADHD, and autism, though our sibling analyzes which better accounted for potential unobserved genetic and environmental confounders showed substantially attenuated associations between valproate and autism which potentially implicates unmeasured confounding. These analyzes need further replication as they are lower powered than traditional analyzes. For example, the risk of intellectual disability in the sibling analysis excluded the null in Sweden (HR: 2.23; 95% CI: 1.01–4.92), but wide uncertainties in the UK pulled down the pooled estimate, resulting in confidence intervals that did not exclude the null (HR: 1.54; 95% CI: 0.76–3.14). We encourage future studies to employ similar and other causal inference methods to better understand the risk of neurodevelopmental outcomes following *in utero* exposure to valproate, as estimates may be inflated by such unmeasured confounding.

## Strength and limitations
Our study has several strengths. We harmonized and combined UK and Swedish registry data. This approach resulted in a large sample size and allowed us to triangulate across two datasets representing two countries and healthcare systems with distinct biases and confounding structures, potentially enhancing the reliability of our findings. Differences in covariates across data sources were observed (e.g., the UK's lower percentage of mothers with no identified indication, 3.4% vs 11.0% among ASM users; the higher percentage of mothers with more than ten healthcare visits in the UK, 59.0% vs 7.2% among ASM users; and Sweden's higher percentage of mothers with neurodevelopmental conditions among ASM users, 17.5% vs 1.41%), and are likely due to the differing nature of the underlying data sources (such as the capture of primary or specialist outpatient care) and resulting definitions used (see Covariates section of the Supplementary Methods). It may be important to note that we observed differences in outcome frequencies, with Sweden having nearly twice the diagnostic prevalence of neurodevelopmental conditions in children, and even more so in mothers, albeit this is consistent with previous studies in the UK[23] and Sweden[24]. Despite these differences, we found comparable effect estimates for most monotherapies (see data source-specific estimates

and tests of heterogeneity in the Supplementary Results). We also compared our primary analyzes to results from methods such as the discordant sibling and active comparator designs, improving our ability to account for time-invariant unobserved confounding. Finally, our study may aid the clinical interpretability of the identified associations, as we estimated both relative and absolute risks across offspring age.

Our study has several limitations. Our results may be affected by measurement error in both the exposure and outcome. For our exposure, we used drug prescription (UK), dispensation and self-reported (Sweden) data rather than the direct assessment of drug consumption (e.g., via serum concentration), meaning that we may have overestimated true exposure during pregnancy. For our outcomes, we relied on diagnoses captured in electronic health records, which may be inaccurate. Reviews of medical records have indicated a high positive predictive value for Swedish data[25], though it has been suggested that neurodevelopmental conditions are under-recorded in CPRD which may explain our lower estimated prevalence of these conditions in the UK[26].

While we have attempted to account for confounding by indication, we are unlikely to have adequately captured confounding by severity of indication, whereby prescription of different ASMs, or combinations of ASMs, may be influenced by how active or severe the indication is, which may also have an impact on risk of neurodevelopmental outcomes. Such confounding is likely to be accounted for in sibling analyzes if it is stable across pregnancies, but not active comparator analyzes. Nonetheless, like any safety analysis on this topic, our study is observational, and there is always a risk that we have not accounted for all relevant confounders or other co-prescriptions. It may be important to recognize the differences between users and non-users of ASMs (Table 1) and consider these when interpreting our findings; even when comparing to lamotrigine users, or siblings to each other, important differences may persist.

Our results may also be affected by the necessary selection upon live births. There is debate about the impact of such selection with some authors suggesting that this will bias results if exposure also influences fetal or neonatal death[27]. Other authors suggest that selection would mean that the interpretation of the estimate is altered to be the effect conditional on livebirth instead of the total effect of in utero ASM exposure on neurodevelopmental outcomes[28].

Although our combined efforts across the UK and Sweden have allowed us to amass one of the largest and longest follow-ups of children diagnosed with neurodevelopmental conditions following ASM exposure in pregnancy, we emphasize that further data on ASM safety are warranted. This need becomes more pertinent as the landscape of ASM continues to evolve, given changing indication prevalences, costs, and known teratogenicity profiles. To enable such large-scale analysis and enhance generalizability, it will remain crucial to involve data from multiple countries across the globe.

## Clinical implications
Clinical recommendations should only be based on the literature as a whole, not individual studies[29]. However, using ASMs during pregnancy requires active monitoring to ensure safety and effectiveness. In cases where pregnant individuals have epilepsy, withdrawal of ASMs can lead to seizures, posing life-threatening risks to both the mother and the baby. Switching to less teratogenic alternatives, such as lamotrigine, well before conception could be a reasonable strategy. Any decisions regarding switching or withdrawal of ASMs should be planned in consultation with relevant clinical specialists and individualized to the patient's clinical history.

The MHRA has stated that the risk of NDD is 30–40% following exposure to valproate[30]. We find substantially lower absolute risks in the UK and Sweden and corroborate results from the Nordic based SCAN-AED study[14,15]. The differences in estimates with those presented

by the MHRA may be explained by our use of electronic health records and registry data with increased sample sizes and lengths of follow up as well as by different measures used to assess neurodevelopment. The difference also highlights the need for better quality evidence and communication of risk in this area.

The findings of this large-scale study suggest that topiramate, carbamazepine, and valproate, but not lamotrigine, use during pregnancy are associated with higher risks of neurodevelopmental conditions in exposed children. These observational associations translate to tangible absolute risk increases (ranging from 0.3 to 2.1 extra cases by age 12 per 100 children exposed *in utero*). These findings may support pregnant women and their physicians to make informed decisions regarding using ASMs during pregnancy.

## Methods

We combined two harmonized cohort studies based on electronic health records from the UK (1995–2018) and Sweden (1995–2020) using meta-analysis. The definitions, processing, and analysis were harmonized across data sources. We developed a common analytical protocol, iterating it until consensus was reached with input from local topic experts. This protocol was then implemented at both sites with only minor revisions to the common protocol (e.g., a broader definition of pain-related indications in Sweden). We pooled the two cohorts using fixed-effects meta-analysis as we could not pool individual-level data because of practical and regulatory constraints. Data source-specific details and a STROBE checklist are provided in the Supplementary Methods, while the overarching methodology and points of departure are described here.

### Data sources

In the UK, we used primary care data from CPRD GOLD with linkage to the CPRD Pregnancy Register, the CPRD Mother-Baby link, the Hospital Episode Statistics (HES) database (including inpatient, outpatient, and emergency care data), Office for National Statistics death certificate data, and Index of Multiple Deprivation (IMD) data. In Sweden, we used the Swedish Developmental Origins of Health and Disease (DOHaD) cohort, which is a registry study linking several national electronic data sources encompassing perinatal care, inpatient care, and specialized outpatient care (from 2005- onwards) throughout the country. Further details on specific data sources at the country level can be found in the Supplementary Methods (Data sources section).

### Study population

In the UK, we included all liveborn children in the CPRD Pregnancy Register born between January 1, 1995, and December 31, 2018 (followed until August 01, 2021) who were linked to maternal information via the CPRD Mother-Baby Link (see Supplementary Methods – Data sources). The CPRD Pregnancy Register lists all pregnancies identified in the CPRD for women aged 11–49 years, with pregnancy episodes identified using an algorithm described in detail by Minassian et al.[31] Briefly, the register includes pregnancy outcome derived from Read codes and Entity types, and estimates of pregnancy timings (including start and end of pregnancy) from date information accompanying the Read codes and Entity types. Validation work, comparing the Pregnancy Register against linked electronic maternity records in HES, has indicated overall good agreement, suggesting most pregnancies are well captured in the register[31]. The CPRD Mother-Baby link allows for linkage between individuals within the same family as a result of a practice specific family identifier[32]. This linkage allowed us to identify prescriptions/covariates in the mothers and diagnoses of neurodevelopmental outcomes in the live-born child. Additionally, we required mothers to be registered at a practice deemed to have continuous high-quality data (referred to as 'up-to-standard' in CPRD documentation[33]) for a minimum of 365 consecutive days before the estimated start of pregnancy, and they needed to remain registered until the expected end of their pregnancy.

In Sweden, we included all liveborn children born between July 1, 1995, and December 31, 2020 (followed until December 31, 2021). Unique personal identifiers were used to link mothers and babies[34], and no selection filter was applied due to the high quality of the linkage. Briefly, the Swedish Medical Birth Register covers 97% to >99% of all deliveries in Sweden between 1973 and 2020[35], as reporting to it is mandatory by all delivering hospitals and care units – the small fraction of missing may be explained by planned or unplanned home deliveries[35] (see Supplementary Methods – Data sources).

The final analytical cohort consisted of 518,047 liveborn children in the UK and 2,664,726 liveborn children in Sweden (see Supplementary Results, Figs. S1 and S2 for flow charts illustrating the cohort derivation process).

### Maternal antiseizure medication

Our primary exposure was fetal exposure to ASM monotherapy at any time during the pregnancy period or polytherapy, defined as exposure to more than one ASM in the pregnancy period. All ASMs were identified using Anatomical Therapeutic Chemical (ATC) codes N03A (ASMs) and N05BA09 (clobazam). In the UK, we identified prescriptions from primary care records, and any pregnancy with a prescription that started or ended during the pregnancy period was classified as exposed. Details of the cleaning procedure are presented in the Supplementary Methods Exposure section (see data availability statement for access to codelists). In Sweden, self-reported maternal consumption of ASMs were prospectively recorded at antenatal care visits (from a structured interview at the first visit at 8–10 weeks of gestation and with recording opportunity at every subsequent visit; the median number of visits in Sweden is ≈9), spanning the period from 1995 to 2020. Beginning the 1st of July 2005, we supplemented this information with drug dispensation data, encompassing all prescription dispensations in Sweden. To account for prescriptions lasting into pregnancy, we also include any prescriptions made up to 30 days before pregnancy. During the period where both self-reported and dispensation data were available, the tetrachoric correlation across the two was 0.98.

### Children's neurodevelopmental conditions

We defined diagnoses of autism, ADHD, and intellectual disability as our primary outcome measures. In the UK, diagnoses were obtained from primary care records using Read codes for all patients and from inpatient and outpatient HES using International Classification of Diseases 10th revision (ICD-10) codes where data linkage was available (details provided in the Supplementary Methods Outcome section). In Sweden, inpatient and specialized outpatient diagnoses were obtained using ICD-8/9/10 codes from the National Patient Registry. To further identify cases of ADHD, we also incorporated prescription information on licensed ADHD medications (recorded using ATC codes at both data sources), including methylphenidate, dexamfetamine, lisdexamfetamine, and atomoxetine. We chose not to include guanfacine due to its second/third-line nature.

Autism is characterized by impairments in social communication and interaction, along with restricted and repetitive behaviours or interests. ADHD involves symptoms in two primary domains: hyperactivity/impulsivity and inattention, with varying extents and presentations. Intellectual disability entails deficits in both intellectual and adaptive functioning, affecting conceptual, social, and practical skills, and typically manifests during the developmental period. Historically, intellectual disability was defined by an intelligence quotient below 70. However, diagnostic criteria have evolved to encompass a more comprehensive assessment, rather than relying solely on intelligence scores. Nonetheless, autism, ADHD and intellectual disability frequently co-occurs.

## Covariates

The confounders were characteristics suspected or known to be associated with prescribing ASMs during pregnancy and with neurodevelopmental conditions in offspring. These covariates encompassed maternal characteristics, including the presumed indication for ASM use (including epilepsy and both psychiatric and somatic conditions), any maternal neurodevelopmental condition diagnoses, maternal age, residential region, evidence of hazardous drinking or illicit drug use during pregnancy, gravidity, maternal health care utilization and seizure events (hospital admission with a seizure ICD-code) in the year before pregnancy, use of antipsychotics and antidepressants in the year before pregnancy, and vomiting (hyperemesis) or antiemetic prescriptions during pregnancy. In the UK, we controlled for IMD quintile, a measure of relative deprivation in small areas, while in Sweden, household disposable income (quintile) and highest household education (categories) were utilized as proxies of socioeconomic position. Detailed definitions of all covariates, which differed between countries due to the availability of data and coding systems employed by each data source and following consultation with local clinical experts, can be found in the Supplementary Methods Covariates section. We further adjusted for year of birth to account for changes in prescribing practices[36] and neurodevelopmental diagnoses[37] over time.

## Primary analysis

We describe the distribution of maternal characteristics for each data source stratified by any ASM exposure status during pregnancy. We estimate the unadjusted and adjusted HR of each offspring neurodevelopmental condition for each ASM monotherapy during pregnancy using flexible parametric survival models[38,39] with three knots for the baseline hazard placed at the 25th, 50th and 75th percentile of the uncensored log survival times, and with cluster-robust standard errors (to account for clustering of children within families) in the UK and Sweden separately. Offspring were followed up from birth, with age as the time scale, to the earliest of diagnosis of a neurodevelopmental condition, end of data collection for the individual, death, or end of follow-up (further details provided in the Supplementary Methods Statistical analysis section). To improve clinical interpretability, we also calculate the absolute adjusted risk and risk difference relative to no ASM, for each outcome assessed at ages 4, 8 and 12 from these models.

We pool all results across data sources using fixed-effects meta-analysis and report these in the main text. Separate UK and Swedish results are reported in the Supplementary Results. We did not use random-effects as only two data sources were included[40].

## Secondary analysis

We performed additional analyses to investigate whether detected associations were potentially causal and clinically relevant. We first employed sibling analyses to account for unobserved genetic and environmental factors shared between siblings[41,42]. In the UK, we identified siblings with the same mother while in Sweden, we identified all full siblings with the same mother and father. In each data source we fit a between-within flexible parametric survival model, estimating a within-family effect of exposure[43]. We then used lamotrigine as an active comparator, typically considered the safest ASM in pregnancy[44]. This may aid clinical interpretability by providing a treatment reference as no ASM use may often not be a realistic alternative for patients prescribed ASM. We used the same models as the main analyses but replaced the reference group. Finally, for each indication, we repeated primary analyses restricted to individuals with the indication. For each analysis we pooled estimates across data sources using fixed-effects meta-analysis.

## Sensitivity analysis

We conducted the following sensitivity analyzes: (1) we ensured a minimum follow-up of 4 years, replicating our primary analysis in cohorts born before 1st of Jan 2018 and 1st of Jan 2017 in Sweden and the UK respectively; (2) we examined exposure in the first trimester, given its relevance to fetal development; (3) we defined exposure by dispensation/prescription on at least two separate occasions during pregnancy (i.e., continued users), using the same definition of polytherapy as in the main analysis. This analysis makes it more likely that a woman uses the same therapeutic throughout a large part of the pregnancy. Finally, (4) recognizing that vomiting or antiemetic prescriptions could act as a mediator occurring after exposure, we repeated primary analyzes excluding this covariate from the adjustment set.

## Tests of heterogeneity

We tested for heterogeneity in hazard ratio estimates between UK and Swedish primary analysis results, between pooled primary analysis and sibling comparison results and between pooled primary analysis and indication restricted analysis results. Results of the active comparator analyzes are necessarily consistent with primary analyzes due to being a rescaling of the model. We used a Wald test, as opposed to $I^2$ statistics to compare coefficients due to the small number of studies included[45].

## Patient and public involvement statement

Patient and public involvement was not employed for this study.

## Ethics approval statement

All research was performed in accordance with relevant guidelines/regulations. The Swedish Ethical Review Authority (DNR 2020-05516) gave ethical approval for the Swedish analysis. Ethics approval for CPRD was obtained from the Independent Scientific Advisory Committee for Medicines and Healthcare products Regulatory Agency database research (Reference: 20_000228). CPRD has obtained ethical approval from a National Research Ethics Service Committee, for all purely observational research using anonymised CPRD data; namely, studies which do not include patient involvement. The Independent Scientific Advisory Committee deemed that this was a non-interventional study and was exempt from the requirement for patient consent as it made use of only linked anonymised data for analysis. Patients were able to opt out of information being shared for research (further details are provided at https://www.cprd.com/public).

## Reporting summary

Further information on research design is available in the Nature Portfolio Reporting Summary linked to this article.

# Data availability

Source data are provided with this paper. Swedish privacy law prohibits us from making registry data publicly available. The data supporting our findings were used under license and ethical approval for the current study. Readers interested in obtaining microdata or replicating our study may seek similar approvals and inquire through Statistics Sweden. For further advice see: https://www.scb.se/en/services/guidance-for-researchers-and-universities/, or contact Statistics Sweden at: mikrodata@scb.se. Access to CPRD data can be obtained by following guidance at https://cprd.com/how-access-cprd-data. Source data are provided with this paper.

# Code availability

Analysis scripts and code lists can be found at https://github.com/pmadleydowd/PREPArE-ASM-Neurodevelopment[46].

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

## Acknowledgements

Many thanks to Dr Hilary Davies-Kershaw for providing Read code lists for illicit drug use and to Dr. Polly Duncan for reviewing code lists for epilepsy. The study was funded by the National Institutes of Health (1R01NS107607: P.M.D., V.H.A., H.F., J.E.R., C.Z., T.T., N.M.D., C.M., D.R. and B.K.L.), Erik and Edith Fernström Foundation for Medical Research (2020-00321: V.H.A.), Karolinska Institutet (2020-00160, 2020-01172: V.H.A.) and the Swedish Society for Medical Research (RM21-0005: V.H.A.). This study was also supported by the NIHR Biomedical Research Centre at the University of Bristol and University Hospitals Bristol and Weston NHS Foundation Trust (NIHR203315: P.M.D., D.R.). P.M.D., F.Z.M., C.J.S.B. and D.R. are members of the UK Medical Research Council (MRC) Integrative Epidemiology unit, which is funded by the MRC (MC_UU_00032/01, MC_UU_00032/02, MC_UU_00032/04 and MC_UU_00032/6) and the University of Bristol. N.M.D. is supported by the Norwegian Research Council (295989). The funders had no role in the design of the study, data collection, data analysis and interpretation of findings.

## Author contributions

D.R., B.K.L. and C.M. conceptualized the study. V.H.A. and P.M.D. performed the formal analysis and drafted the original manuscript. D.R., B.K.L. and C.M. supervised the study. B.K.L., K.L., C.N., D.R., N.M.D., C.M. and T.T. were involved in funding acquisition. M.L. performed data curation of Swedish registries. P.M.D., J.E.R. and H.F. performed data curation of UK CPRD GOLD data. P.M.D., V.H.A., H.F., J.E.R., F.Z.M., C.Z., C.J.S.B., D.B, M.L, K.L, C.J.N., T.T., N.M.D., C.M., D.R. and B.K.L. critically revised the manuscript for important intellectual content and contributed to the development of the study methodology. V.H.A. and P.M.D. had full access to all the data in the study and take responsibility for the integrity of the data and the accuracy of the data analysis.

## Funding

## Competing interests

T.T. has received support to the EURAP pregnancy registry from Accord, Eisai, GSK, UCB, Bial, Sanofi, GW Pharma, Teva, Angelini Pharma, Zentiva, SF Group, Glenmark and personal fees from Eisai, Sanofi, UCB and Angelini Pharma outside the submitted work. BL reports receiving consulting fees from Beasley Allen Law Firm, Patterson Belknap Webb & Tyler LLP and AlphaSights. None of the above entities had any role in the design and conduct of the study; collection, management, analysis and interpretation of the data; preparation, review, or approval of the manuscript; and decision to submit the manuscript for publication. The remaining authors have no conflicts of interest.
