## [Transparent Peer Review file · Nature Communications]

Antiseizure Medication Use During Pregnancy and Children's Neurodevelopmental Outcomes

Corresponding Author: Dr Viktor Ahlqvist

Version 0:

Reviewer comments:

Reviewer #1

(Remarks to the Author)

1. Line 82, authors stated that there is a small number of pregnancies exposed to topiramate in previous studies, and the one in the US has N=623. However, this present study only has N=418, in total, exposed to topiramate as well with small number of events in both studies and thus it does not seem the current study has a much larger sample in that sense. However, given the event rate for neurodevelopmental outcomes is roughly around 4 to 8% in general and highly depending on the age catchment, not only the number of individuals but also how old are the participants and the length of follow-up are no less important.
2. Line 95, authors may consider adding "time-invariant" before "unobserved confounding". Sibling-matched analysis is only able to address shared familial confounding that is constant over time. It cannot address all unobserved confounding.
3. Line 99, the current study is essentially consisted with two cohort studies. The authors mentioned the two cohort studies were harmonised in terms of "definitions, processing, and analysis" across the two data sources. It would be helpful to have more details on how and what did the investigators do in harmonising the two data sources: is it based on a common analytic protocol or based on existing common data model? Either way it would be informative to the readers about the code mapping/translation between the two data source given that their coding system on both conditions and medications are not the same.
4. Line 112, has the registry DOHaD in Sweden been validated or adopted in other previous studies? Please share some previous studies using the same registry in the Supplementary information.
5. Line 115, 175 and 259, authors may consider specifying the part/ section within the supplementary methods. Consider numbering/ labeling different sections as this will make the reader easier to follow.
6. Line 118 and 124, the shortest follow up will be 2.5 years in the UK cohort and the shortest follow up for the Sweden cohort will be 1 year. That refers back to the my first comment that the length of follow-up matters for some outcomes e.g. ADHD and ID. ADHD is typically being diagnosed after 5 years old. Please further elaborate on the censoring criteria of the follow-up period.
7. Line 127, it would be good to mention this study only live birth in the method section. Also, it would be helpful to move some of the information in the supp to the main method in particular for how pregnancy periods were identified in each database.
8. Line 129, does exposure have to be prescription or dispensation records during pregnancy or can it be something like having a prescription before pregnancy but the expected prescription end during pregnancy?
9. Line 141, further dividing the timing of exposure into trimesters may offer a more comprehensive clinical insight on the impact of timing of antiseizure medications. Exposure in the first trimester may have different impact towards the child, correspond to the development phase of the foetus, when exposed in second or third trimester. At the same time, pregnancy may disrupt the treatment because of the safety concerns of the medication.

10. Line 150, guanfacine is also licensed for ADHD in both countries. If exposure to guanfacine was not used to define ADHD cases, please kindly justify.
11. Line 159, have the authors considered also including other medications such as lithium, ADHD medications, anxiolytics, antihypertensives, steroids, antibiotics and opioids as well? They can be potential confounders in this study.
12. Line 160, vomiting or antiemetic prescriptions during pregnancy were listed as covariates and in theory these could occurred after the occurrence of the exposure and thus could potentially mediated the results.
13. Line 163, the supplementary table quoted does not describe the covariates.
14. Line 181, to account for potential exposure misclassification, have the authors considered limiting the exposed cohort to patients who have at least 2 prescriptions as one of the secondary analyses?
15. The difference between the exposed and the unexposed groups were clearly demonstrated in Table 1. Some of these differences may lead to residual confounding even after adjustment in the analysis and in the sibling analysis. Propensity score is generally applied not only to address confounding but also to demonstrate if the exposure groups are comparable. It would be helpful if the investigator could discuss the comparability between the groups. Authors may also consider adding mental health comorbidities at baseline in Table 1.
16. Typo in Table 1, calendar year of delivery, 2019-20 for the unexposed group in Sweden. It should be 245890.
17. In table S10, it is noted that the limited number of patients were identified for some of the outcomes for individual antiseizure medications. It may worth addressing that in the limitation section under discussion.
18. Line 225, the phrase "in addition" was repeated.
19. Line 234-235, table should be S8 instead of S7.
20. Line 250, as stated in the supplementary information, why is neuropathic pain not included as somatic indications in the Swedish cohort? There are discrepancies for the somatic indications between the UK and Swedish cohort. It will be much appreciated if the authors can explain such difference.
21. Line 286-287, according to Supplementary Table S13, some of the HRs of Carbamazepine associated with autism and ID are not significantly larger than 1 when analysis was stratified by different indications. This contradicts to this statement.
22. Line 296, the main analysis also shows that there is a higher risk for autism.
23. Line 297-298, the HR for ID was also attenuated and confidence intervals cover 1, should the authors consider mentioning it in this statement as well?

Reviewer #2

(Remarks to the Author)

Thank you for the opportunity to review this paper. The study provides valuable insights into the impact of antiseizure medication use during pregnancy on children's neurodevelopmental outcomes, utilizing robust data from electronic health records in the UK and Sweden. Please kindly find some of the comments that may benefit the clinical implication of this paper.

1. Line 82, authors stated that there is a small number of pregnancies exposed to topiramate in previous studies, and the one in the US has N=623. However, this present study only has N=418, in total, exposed to topiramate as well with small number of events in both studies and thus it does not seem the current study has a much larger sample in that sense. However, given the event rate for neurodevelopmental outcomes is roughly around 4 to 8% in general and highly depending on the age catchment, not only the number of individuals but also how old are the participants and the length of follow-up are no less important.
2. Line 95, authors may consider adding "time-invariant" before "unobserved confounding". Sibling-matched analysis is only able to address shared familial confounding that is constant over time. It cannot address all unobserved confounding.
3. Line 99, the current study is essentially consisted with two cohort studies. The authors mentioned the two cohort studies were harmonised in terms of "definitions, processing, and analysis" across the two data sources. It would be helpful to have more details on how and what did the investigators do in harmonising the two data sources: is it based on a common analytic protocol or based on existing common data model? Either way it would be informative to the readers about the code mapping/translation between the two data source given that their coding system on both conditions and medications are not the same.
4. Line 112, has the registry DOHaD in Sweden been validated or adopted in other previous studies? Please share some previous studies using the same registry in the Supplementary information.

5. Line 115, 175 and 259, authors may consider specifying the part/ section within the supplementary methods. Consider numbering/ labeling different sections as this will make the reader easier to follow.
6. Line 118 and 124, the shortest follow up will be 2.5 years in the UK cohort and the shortest follow up for the Sweden cohort will be 1 year. That refers back to the my first comment that the length of follow-up matters for some outcomes e.g. ADHD and ID. ADHD is typically being diagnosed after 5 years old. Please further elaborate on the censoring criteria of the follow-up period.
7. Line 127, it would be good to mention this study only live birth in the method section. Also, it would be helpful to move some of the information in the supp to the main method in particular for how pregnancy periods were identified in each database.
8. Line 129, does exposure have to be prescription or dispensation records during pregnancy or can it be something like having a prescription before pregnancy but the expected prescription end during pregnancy?
9. Line 141, further dividing the timing of exposure into trimesters may offer a more comprehensive clinical insight on the impact of timing of antiseizure medications. Exposure in the first trimester may have different impact towards the child, correspond to the development phase of the foetus, when exposed in second or third trimester. At the same time, pregnancy may disrupt the treatment because of the safety concerns of the medication.
10. Line 150, guanfacine is also licensed for ADHD in both countries. If exposure to guanfacine was not used to define ADHD cases, please kindly justify.
11. Line 159, have the authors considered also including other medications such as lithium, ADHD medications, anxiolytics, antihypertensives, steroids, antibiotics and opioids as well? They can be potential confounders in this study.
12. Line 160, vomiting or antiemetic prescriptions during pregnancy were listed as covariates and in theory these could occurred after the occurrence of the exposure and thus could potentially mediated the results.
13. Line 163, the supplementary table quoted does not describe the covariates.
14. Line 181, to account for potential exposure misclassification, have the authors considered limiting the exposed cohort to patients who have at least 2 prescriptions as one of the secondary analyses?
15. The difference between the exposed and the unexposed groups were clearly demonstrated in Table 1. Some of these differences may lead to residual confounding even after adjustment in the analysis and in the sibling analysis. Propensity score is generally applied not only to address confounding but also to demonstrate if the exposure groups are comparable. It would be helpful if the investigator could discuss the comparability between the groups. Authors may also consider adding mental health comorbidities at baseline in Table 1.
16. Typo in Table 1, calendar year of delivery, 2019-20 for the unexposed group in Sweden. It should be 245890.
17. In table S10, it is noted that the limited number of patients were identified for some of the outcomes for individual antiseizure medications. It may worth addressing that in the limitation section under discussion.
18. Line 225, the phrase "in addition" was repeated.
19. Line 234-235, table should be S8 instead of S7.
20. Line 250, as stated in the supplementary information, why is neuropathic pain not included as somatic indications in the Swedish cohort? There are discrepancies for the somatic indications between the UK and Swedish cohort. It will be much appreciated if the authors can explain such difference.
21. Line 286-287, according to Supplementary Table S13, some of the HRs of Carbamazepine associated with autism and ID are not significantly larger than 1 when analysis was stratified by different indications. This contradicts to this statement.
22. Line 296, the main analysis also shows that there is a higher risk for autism.
23. Line 297-298, the HR for ID was also attenuated and confidence intervals cover 1, should the authors consider mentioning it in this statement as well?

Reviewer #3

(Remarks to the Author)

This is a substantial study of UK and Swedish data investigating exposures to ASM and outcomes of autism, ADHD and intellectual disability. The study includes full populations from two electronic healthcare databases of women with pregnancies ending in live births that include the years 1995-2018 (or 2020 for Sweden). Mothers needed 1 year of data

before the start of their pregnancy and to be followed up until the end of their pregnancy. Analyses of ASM exposed versus unexposed in the full population and secondary analyses of siblings and using lamotrigine as an active comparator are included. An extensive set of results are presented in the supplementary information supplied by the authors.

It is appreciated that word counts can limit the inclusion of all details that might be needed to fully explain the work presented and the supplementary information can enable more to be included, however some further clarification of the following would be helpful in the main paper:

Follow up: Please comment on whether there is any impact of including children born in 2020 but only followed up for 1 year (Sweden data) given that it is unlikely that the outcomes of interest could be detected by one year of age (apart from very profound developmental delay). Would a minimum follow up time for the children born following the included pregnancies give more potential for diagnoses of the outcomes of interest e.g. child development checks at pre-school ages?

Exposure: Please could you define how exposure is classified during the pregnancy. Is this by receipt of any ASM at any time or is this broken down by trimester or month of pregnancy? Are changes in exposures during pregnancy accounted for in the analyses or did women use the same medication throughout? How is polytherapy defined?

Outcome: Please define intellectual disability further - does this include learning difficulty and/ or learning disability and /or IQ scores? Please also state whether the prevalence of the outcomes achieved is as expected in the two populations and if these are not as expected then suggest why this is the case. The supplementary file indicates that the numbers with ID in the UK are very low (S2) so more explanation about whether cases are missed or not in the presentation of these results is needed.

Results: it was noted that diagnoses of mothers with NDD is much higher in Sweden than in the UK. Why is this? Are diagnostic criteria or methods of case identification different?

Supplementary file: Please could you add footnotes to state what the counts of outcomes represent - whether this is at 4, 8 or 12 years or for the full study. Is there scope to include aspects of S10 and S13 in the main paper since there is little published about some of these exposures and these results may prove helpful to those contemplating pregnancy and their healthcare workers. However some editing would be needed to highlight the channelling of some medications towards one set of maternal indications over others e.g. pregabalin or gabapentin are unlikely to be prescribed for epilepsy.

Extended figure 1: Could the epilepsy only data for this study in the figure at the relevant points to give this comparison?

Overall, this is an extensive piece of work that is well described and will be of significance in the field. The challenge in presenting this work though is that there is a lot of detail to be described and data that is needed to fully understand what has been found to enable this to be put into context with others' work. I would prefer to see extra supplementary tables from the results given in the main paper and the extended data table comparison (page 24) (while interesting to review) to be included in the supplementary file.

Version 1:

Reviewer comments:

Reviewer #1

(Remarks to the Author)

We would like to thank the research team for their careful attention to our comments. The updated version addresses some of the points raised in the initial review. However, we have a few additional suggestions that will further enhance the robustness of the study.

1. In response to the first comment, we acknowledge that the authors have revised the sentence in Lines 83-87. While it is true that the current study may have an advantage in follow-up time and sample size for Topiramate, the overall cohort, which includes various ASMs, has a smaller sample size and shorter follow-up time compared to the referenced Nordic study.

2. In response to the second comment, in the Supplementary Methods (pages 9-13), it is not only the definition of neuropathic pain that differs. It would be clearer for readers if it were explicitly noted that the definitions of covariates and comorbidities also vary between the two databases, following consultation with local experts. Additionally, it would be helpful to explain to readers why these differences exist and whether they would affect the interpretation of the results.

3. In response to comment 5, Lines 122 and 299 still do not specify the exact parts or sections of the supplementary materials. For example, in Lines 298-299, the statement "Drug-specific estimates are reported in Table 2 and the Supplementary Results" should specify which particular table in the Supplementary Results. Similarly, in Lines 121-122, where it states, "Further details on specific data sources at the county level can be found in the Supplementary Methods," please indicate the specific section of the Supplementary Methods.

4. In response to comments 6 and 9, it is notable that valproate no longer shows a significantly increased risk of ADHD after restricting follow-up time to at least 4 years or when focusing on first-trimester exposure. The authors may want to highlight this finding to suggest that interpretation should be approached with caution. Since ADHD typically requires around 5 years for diagnosis, the shorter follow-up time in this study should also be addressed in the limitations section.

5. In response to comment 11, the authors stated that "While some of the drugs mentioned by the reviewer may influence the risk of neurodevelopmental conditions in children, it is difficult to envision them having a causal effect on the use of antiseizure medications. Therefore, we did not consider including these additional medications in our analysis; without an effect on the exposure, they would not lead to confounding." Please kindly share references to justify such claim.

If the investigator "believe" it is difficult to envision, they have to provide evidence to support such believe. The relationship between a potential confounding factor and the exposure can be simply a correlation, not necessarily a causal factor, to confound the outcome estimate.

6. In response to comment 12, please consider presenting the results in supplementary results.

7. In response to comment 17, the authors revised the final paragraph under strengths and limitations, "Although our combined efforts across the UK and Sweden have allowed us to amass one of the largest and longest follow-ups of children diagnosed with neurodevelopmental conditions following ASM exposure in pregnancy, we emphasize that further data on ASM safety are warranted. This need becomes more pertinent as the landscape of ASM continues to evolve, given changing indication prevalences, costs, and known teratogenicity profiles. To enable such large-scale analysis and enhance generalizability, it will remain crucial to involve data from multiple countries across the globe." The highlighted part is not true, compared to the quoted Nordic study, which has a larger sample size and longer follow-up time.

8. In response to comment 20, if the Swedish registry uses a broader definition for somatic conditions, why was this definition not applied to the UK database? Additionally, it's important to note that the Swedish database contributed a substantial weight to the majority of the analyses. This discrepancy may warrant mentioning in the manuscript.

9. It is recommended to add footnote, explaining all the abbreviations used in the all tables, including ones in the supplementary results.

Reviewer #2

(Remarks to the Author)

Reviewer #3

(Remarks to the Author)

Many thanks to the authors for their thorough review and response to all of the comments submitted in the first review stage. The paper is very detailed in its methods and presentation of results and reads clearly and well. The revisions have mostly addressed the areas that concerned me but the following remain as points that could be updated or further clarified in the text:

Intellectual disability: thank you for including the definition of this diagnosis. As the definition of this has changed over time to not just include IQ scores, please would you also include the code lists (or a link to a repository holding these) to aid others in comparing or undertaking to report similar work.

Polytherapy: I recognise that many combinations of ASMs may be present in the data; given the findings for valproate, carbamazepine and topiramate, please could you indicate if any of the results for polytherapy include any of these three ASMs? This is very important to understand in terms of the potential influence that these three ASMs could be having on the polytherapy results.

Pg14, line 410: the data from the MHRA review is based on field studies rather than database studies. This highlights the difficulty that database studies can have in capturing all outcomes of this type - sometimes because people are lost to follow up or diagnoses take a long time to be made. You have noted this already in a previous paragraph therefore an update to indicate the differences in the types of studies and results found would help to explain this disparity.

REVIEWER COMMENTS

Reviewer #1 & #2 (Remarks to the Author):

Thank you for the opportunity to review this paper. The study provides valuable insights into the impact of antiseizure medication use during pregnancy on children's neurodevelopmental outcomes, utilizing robust data from electronic health records in the UK and Sweden. Please kindly find some of the comments that may benefit the clinical implication of this paper.

Answer: Thank you for your insightful comments and acknowledgment of our work. Your detailed and thorough feedback has greatly contributed to enhancing the quality of our manuscript, resulting in a more comprehensive description of our research.

We would also like to express our gratitude to both the early career researchers and their mentors for their collaborative efforts in the review process. We believe that mentorship programs like this one are invaluable initiatives that foster growth and development within the scientific community.

1. Line 82, authors stated that there is a small number of pregnancies exposed to topiramate in previous studies, and the one in the US has N=623. However, this present study only has N=418, in total, exposed to topiramate as well with small number of events in both studies and thus it does not seem the current study has a much larger sample in that sense. However, given the event rate for neurodevelopmental outcomes is roughly around 4 to 8% in general and highly depending on the age catchment, not only the number of individuals but also how old are the participants and the length of follow-up are no less important.

Answer: Thank you for allowing us to clarify this point. The relevant section refers both to the prior Nordic study which had fewer monotherapy exposed pregnancies (N=246 in SCAD-AED vs N=418 in our study), and the US which had relatively few exposed cases (N<11 monotherapy exposed autism cases, actual number not reported due to the small count). As the reviewer alludes to, the US study was especially limited in its follow-up of children's outcomes (with a median follow-up of 2 years). This limitation likely stems from the study's reliance on insurance claim systems for follow-up, where children are lost if they are no longer covered by the same insurance provider, which often occurs when a parent changes employer or health care provider.

We have now revised the relevant section to include a note about the importance of longer follow-up of children (third paragraph of the background):

“However, the small number of pregnancies exposed to monotherapy topiramate (N=246 in the Nordic¹⁵ & N=623 in the US¹⁸), the few cases of neurodevelopmental outcomes (N=10 in the Nordic¹⁵ & N<11 in the US¹⁸), and the short follow-up of the children (median 5.7

years in the Nordic¹⁵ & 2 years in the US study¹⁸), mean more evidence is needed from large populations with longitudinal follow up about the risks associated with these medications.”

2. Line 95, authors may consider adding “time-invariant” before “unobserved confounding”. Sibling-matched analysis is only able to address shared familial confounding that is constant over time. It cannot address all unobserved confounding.

Answer: We agree that this is important to recognize and have now amended it throughout the manuscript (in the final paragraph of background and in the first paragraph of the strengths and limitations section of the discussion,).

3. Line 99, the current study is essentially consisted with two cohort studies. The authors mentioned the two cohort studies were harmonised in terms of “definitions, processing, and analysis” across the two data sources. It would be helpful to have more details on how and what did the investigators do in harmonising the two data sources: is it based on a common analytic protocol or based on existing common data model? Either way it would be informative to the readers about the code mapping/translation between the two data source given that their coding system on both conditions and medications are not the same.

Answer: Thank you for allowing us to clarify this. We developed a common analytical protocol, led by PMD and VHA, which was circulated among the author group, comprising experts in pregnancy drug safety, antiseizure teratology and epilepsy, neurodevelopmental conditions, and statistics. This process was iterated until consensus on the analytical protocol was reached. To harmonize the definitions of diagnostic codes and drug codes, we consulted local experts to ensure consistent data capture across the countries (TT & CM in Sweden, and as described in the acknowledgement, Dr Hilary Davies-Kershaw and Dr Polly Duncan in the UK).

During the implementation of the final protocol and coding, several group members met on a weekly basis, and we had to make minor revisions. For example, we discovered insufficient resolution to detect specific neuropathic pains in Sweden, leading to a broader definition (see reply to comment below for details). The protocol was revised accordingly. The final protocol is largely what became the supplemental methods for the original submission.

We have now revised the first paragraph of the method section and clarified the iteration-until-consensus process:

“The definitions, processing, and analysis were harmonised across data sources. We developed a common analytical protocol, iterating it until consensus was reached with input from local topic experts. This protocol was then implemented at both sites with only minor revisions to the common protocol (e.g., a broader definition of pain-related indications in

Sweden). We pooled the two cohorts using fixed-effects meta-analysis, as we could not pool individual-level data because of practical and regulatory constraints.”

4. Line 112, has the registry DOHaD in Sweden been validated or adopted in other previous studies? Please share some previous studies using the same registry in the Supplementary information.

Answer: Thank you for allowing us to clarify this. DOHaD is a study/database that integrates several nationwide and regional Swedish registries to facilitate evaluations of pregnancy drug safety and practice. It is not a registry itself and thus does not require separate validation; it relies on the validated, high-quality data from the underlying Swedish registries. All contributing registries have been validated, thoroughly described, and are widely used in the literature. In the original submission supplementary methods, we referred to detailed content and validity reports on each used registry. However, we recognize that in the original submission it was not entirely clear how DOHaD relates to these registries.

We now provide a more detailed explanation of DOHaD in the supplementary information. Additionally, as a partial validation of its use in drug safety studies, we now also refer to a recent publication wherein we used DOHaD to study acetaminophen safety in pregnancy (published in JAMA 2024, PMID: 38592388).

5. Line 115, 175 and 259, authors may consider specifying the part/ section within the supplementary methods. Consider numbering/ labeling different sections as this will make the reader easier to follow.

Answer: Thank you for this valuable suggestion. We have now revised this throughout the main article; at each point where we refer to the supplement, we now guide readers to specific sections.

6. Line 118 and 124, the shortest follow up will be 2.5 years in the UK cohort and the shortest follow up for the Sweden cohort will be 1 year. That refers back to the my first comment that the length of follow-up matters for some outcomes e.g. ADHD and ID. ADHD is typically being diagnosed after 5 years old. Please further elaborate on the censoring criteria of the follow-up period.

Answer: Thank you for allowing us to elaborate on this matter, as we agree that it is important. As described on line 172 of the original submission, children “... were followed up from birth, with age as the time scale, to the earliest of diagnosis of a neurodevelopmental condition, end of data collection for the individual, death, or end of follow up”.

It is true that the *minimum* follow-up was 2.5 years in the UK and 1 year in Sweden, and few children are diagnosed so early. However, this represents a relatively minor part of the cohorts.

We direct the reviewer to Supplementary Table S1, where we show the median follow-up times for the UK (7.5 years) and Sweden (12.5 years) separately. Our longer median follow-up of children likely explains why we have been able to surpass prior studies, especially those from the Nordic countries, in the number of children diagnosed with neurodevelopmental conditions. It also distinguishes us from the US study of autism, which had a median follow-up of 2 years.

To accommodate this and other reviewers' interest in an analysis with a minimum follow-up period until pre-school ages, where developmental checks are typically performed, we have now conducted a complementary analysis that includes only children born before 1st of Jan 2018 and 1st of Jan 2017 in Sweden and UK respectively, ensuring a minimum of four years of follow-up for all children. This analysis shows similar findings to our main analysis. We have briefly commented on this in the methods section (sensitivity analyses) and results section (sensitivity analyses and new supplementary table S13).

Methods: “We conducted the following sensitivity analyses: 1) we ensured a minimum follow-up of 4 years, replicating our primary analysis in cohorts born before 1st of Jan 2018 and 1st of Jan 2017 in Sweden and the UK respectively; ...”

Results: “Results of sensitivity analyses 1-3 are presented in supplementary Table S13. Restricting the cohorts to ensure a minimum follow-up of 4 years did not change the conclusions of our primary analyses. ...”

For reviewer reference, below is an excerpt of the table summarizing the results of this analysis:

		HR (95% CI)		Weight for Sweden	
		Primary analysis	Restricted cohort analysis	Primary analysis	Restricted cohort analysis
ASD	No ASM	1 (Reference)	1 (Reference)	-	-
	Carbamazepine	1.25 (1.05-1.48)	1.17 (0.98-1.38)	88.0%	88.1%
	Gabapentin	1.15 (0.84-1.59)	1.23 (0.89-1.69)	56.0%	56.0%
	Lamotrigine	0.86 (0.72-1.01)	0.91 (0.77-1.08)	88.1%	88.0%
	Levetiracetam	0.82 (0.44-1.54)	1.02 (0.54-1.90)	80.3%	80.4%
	Phenytoin	0.99 (0.55-1.80)	0.84 (0.47-1.52)	90.3%	90.3%
	Pregabalin	0.71 (0.54-0.93)	0.70 (0.53-0.92)	87.1%	92.1%
	Topiramate	1.14 (0.68-1.93)	1.04 (0.60-1.80)	92.9%	92.4%
	Valproate	1.78 (1.48-2.14)	1.67 (1.39-2.00)	89.0%	89.1%
	Other	1.30 (0.93-1.84)	1.20 (0.85-1.69)	91.0%	91.0%
	Polytherapy	1.51 (1.22-1.88)	1.50 (1.20-1.86)	81.2%	81.7%
ID	No ASM	1 (Reference)	1 (Reference)	-	-
	Carbamazepine	1.30 (1.01-1.69)	1.26 (0.97-1.63)	94.0%	94.0%
	Gabapentin	1.02 (0.49-2.15)	0.77 (0.32-1.85)	100.0%	100.0%
	Lamotrigine	1.12 (0.86-1.46)	1.18 (0.91-1.53)	93.4%	93.4%
	Levetiracetam	0.66 (0.21-2.05)	0.77 (0.25-2.41)	100.0%	100.0%
	Phenytoin	0.67 (0.22-2.10)	0.62 (0.20-1.93)	100.0%	100.0%
	Pregabalin	0.86 (0.52-1.40)	0.89 (0.55-1.47)	100.0%	100.0%
	Topiramate	2.48 (1.23-4.98)	2.55 (1.27-5.12)	87.7%	87.7%
	Valproate	2.56 (1.97-3.32)	2.42 (1.86-3.14)	96.7%	96.7%
	Other	0.90 (0.45-1.81)	0.88 (0.44-1.77)	100.0%	100.0%

	Polytherapy	2.02 (1.49-2.75)	2.07 (1.52-2.81)	91.1%	91.1%
ADHD	No ASM	1 (Reference)	1 (Reference)	-	-
	Carbamazepine	1.07 (0.94-1.21)	0.99 (0.87-1.12)	94.6%	94.7%
	Gabapentin	1.22 (0.92-1.63)	1.24 (0.94-1.65)	81.5%	81.5%
	Lamotrigine	0.89 (0.78-1.01)	0.95 (0.83-1.08)	94.8%	94.8%
	Levetiracetam	1.05 (0.64-1.74)	1.23 (0.74-2.03)	90.9%	90.8%
	Phenytoin	1.16 (0.79-1.70)	0.96 (0.66-1.41)	96.3%	96.3%
	Pregabalin	0.67 (0.55-0.83)	0.70 (0.57-0.86)	96.8%	97.8%
	Topiramate	0.80 (0.49-1.30)	0.82 (0.50-1.33)	94.0%	94.0%
	Valproate	1.20 (1.02-1.40)	1.11 (0.95-1.29)	95.5%	95.5%
	Other	0.95 (0.71-1.25)	0.87 (0.66-1.16)	98.0%	98.0%
	Polytherapy	1.12 (0.92-1.36)	1.12 (0.92-1.36)	93.8%	93.8%

7. Line 127, it would be good to mention this study only live birth in the method section. Also, it would be helpful to move some of the information in the supp to the main method in particular for how pregnancy periods were identified in each database.

Answer: Thank you for bringing up this important point. While this is reflected in the two paragraphs above line 127, we have now also revised the relevant sentence to reflect that only liveborn children are included (final paragraph of the study population section in the methods of the revised manuscript):

“The final analytical cohort consisted of 518,047 liveborn children in the UK and 2,664,726 liveborn children in Sweden”

Furthermore, we have now added information to the main article from the supplemental material about how pregnancy periods were identified in each database. For the UK (first paragraph of the study population section of the methods):

“In the UK, we included all liveborn children in the CPRD Pregnancy Register born between January 1, 1995, and December 31, 2018 (followed until August 01, 2021) who were linked to maternal information via the CPRD Mother-Baby Link (see *Supplemental Methods – Data sources*). The CPRD Pregnancy Register lists all pregnancies identified in the CPRD for women aged 11–49 years, with pregnancy episodes identified using an algorithm described in detail by Minassian et al.⁴ Briefly, the register includes pregnancy outcome derived from Read codes and Entity types, and estimates of pregnancy timings (including start and end of pregnancy) from date information accompanying the Read codes and Entity types. Validation work, comparing the Pregnancy Register against linked electronic maternity records in HES, has indicated overall good agreement, suggesting most pregnancies are well captured in the register.⁴ The CPRD Mother-Baby link allows for linkage between individuals within the same family as a result of a practice specific family identifier⁵. This linkage al-

lowed us to identify prescriptions/covariates in the mothers and diagnoses of neurodevelopmental outcomes in the live-born child. Additionally, we required mothers to be registered at a practice deemed to have continuous high-quality data (referred to as 'up-to-standard' in CPRD documentation²²) for a minimum of 365 consecutive days before the estimated start of pregnancy, and they needed to remain registered until the expected end of their pregnancy.”

And for Sweden (second paragraph of the study population section of the methods):

“In Sweden, we included all liveborn children born between July 1, 1995, and December 31, 2020 (followed until December 31, 2021). Unique personal identifiers were used to link mothers and babies²³, and no selection filter was applied due to the high quality of the linkage. Briefly, the Swedish Medical Birth Register covers 97% to >99% of all deliveries in Sweden between 1973 and 2020²⁴, as reporting to it is mandatory by all delivering hospitals and care units – the small fraction of missing may be explained by planned or unplanned home deliveries²⁴ (see *Supplemental Methods – Data sources*).”

8. Line 129, does exposure have to be prescription or dispensation records during pregnancy or can it be something like having a prescription before pregnancy but the expected prescription end during pregnancy?

Answer: Thank you for allowing us to clarify this matter. The overarching information is available in the original main article, with further details provided in the supplement.

In Sweden, a pregnancy was classified as exposed if the mother reported antiseizure medication intake at any antenatal visit (i.e., during pregnancy). After July 1st, 2005, this classification was supplemented by considering whether she dispensed antiseizure medications within 30 days before the start of pregnancy up until the delivery date. Thanks to the reviewer, we noticed that we had failed to describe this 30-day buffer of drug dispensation data in the original submission, this has now been revised accordingly in the main article (excerpt below) and the supplemental material (Supplemental Methods – Data Sources - DOHaD).

In the UK, any prescription spanning the pregnancy period contributed to exposure classification. Thus, a pregnancy with a prescription that started before but ended during pregnancy would be classified as exposed.

To further aid readers, we believe that certain details can be relocated from the supplementary material and elaborated on in the main article. We have now revised the exposure definition section to read:

“Our primary exposure was fetal exposure to ASM monotherapy at any time during the pregnancy period or polytherapy, defined as exposure to more than one ASM in the pregnancy period. All ASMs were identified using Anatomical Therapeutic Chemical (ATC) codes N03A (antiseizure medications) and N05BA09 (clobazam). In the UK, we identified prescriptions from primary care records, and any pregnancy with a prescription that started or ended during the pregnancy period was classified as exposed. Details of the cleaning procedure are presented in the Supplementary Methods (see data availability statement for access to codelists). In Sweden, self-reported maternal consumption of ASMs were prospectively recorded at antenatal care visits (from a structured interview at the first visit at 8-10 weeks of gestation and with recording opportunity at every subsequent visit; the median number of visits in Sweden is ≈9), spanning the period from 1995 to 2020. Beginning the 1st of July 2005, we supplemented this information with drug dispensation data, encompassing all prescription dispensations in Sweden. To account for prescriptions lasting into pregnancy, we also include any prescriptions made up to 30 days before pregnancy. During the period where both self-reported and dispensation data were available, the tetrachoric correlation across the two was 0.98.”

9. Line 141, further dividing the timing of exposure into trimesters may offer a more comprehensive clinical insight on the impact of timing of antiseizure medications. Exposure in the first trimester may have different impact towards the child, correspond to the development phase of the foetus, when exposed in second or third trimester. At the same time, pregnancy may disrupt the treatment because of the safety concerns of the medication.

Answer: We appreciate the reviewers’ interest in this important matter. We concur with the reviewer that exposure during the first trimester is of particular interest due to the developmental phase of the fetus. Therefore, we have decided to conduct additional analyses on first-trimester exposure. The result of this analysis suggests that results are broadly comparable to the primary analyses for all ASM and NDD combinations with the exception of phenytoin and ADHD (phenytoin is a well-known teratogen and typically already contraindicated during pregnancy). We have now included comments on this analysis in the methods and results section in subsections titled “Sensitivity analysis”.

For reviewer reference, below is an excerpt of the table summarizing the results of this analysis:

		HR (95% CI)		Weight for Sweden	
		Primary analysis	Trimester 1 analysis	Primary analysis	Trimester 1 analysis
ASD	No ASM	1 (Reference)	1 (Reference)	-	-
	Carbamazepine	1.25 (1.05-1.48)	1.35 (1.03-1.77)	88.0%	72.1%
	Gabapentin	1.15 (0.84-1.59)	1.19 (0.82-1.71)	56.0%	45.6%
	Lamotrigine	0.86 (0.72-1.01)	0.92 (0.76-1.11)	88.1%	85.7%
	Levetiracetam	0.82 (0.44-1.54)	1.27 (0.70-2.32)	80.3%	82.0%
	Phenytoin	0.99 (0.55-1.80)	1.19 (0.18-8.05)	90.3%	0.0%

	Pregabalin	0.71 (0.54-0.93)	0.79 (0.60-1.05)	87.1%	86.3%
	Topiramate	1.14 (0.68-1.93)	1.33 (0.79-2.26)	92.9%	92.9%
	Valproate	1.78 (1.48-2.14)	1.61 (1.20-2.15)	89.0%	71.2%
	Other	1.30 (0.93-1.84)	1.34 (0.84-2.13)	91.0%	83.5%
	Polytherapy	1.51 (1.22-1.88)	1.52 (1.16-2.00)	81.2%	73.1%
ID	No ASM	1 (Reference)	1 (Reference)	-	-
	Carbamazepine	1.30 (1.01-1.69)	1.36 (0.87-2.12)	94.0%	87.0%
	Gabapentin	1.02 (0.49-2.15)	1.11 (0.46-2.66)	100.0%	100.0%
	Lamotrigine	1.12 (0.86-1.46)	1.12 (0.82-1.54)	93.4%	90.7%
	Levetiracetam	0.66 (0.21-2.05)	1.26 (0.46-3.40)	100.0%	100.0%
	Phenytoin	0.67 (0.22-2.10)	2.77 (0.39-19.84)	100.0%	100.0%
	Pregabalin	0.86 (0.52-1.40)	0.92 (0.54-1.57)	100.0%	100.0%
	Topiramate	2.48 (1.23-4.98)	2.47 (1.17-5.21)	87.7%	86.0%
	Valproate	2.56 (1.97-3.32)	2.06 (1.29-3.29)	96.7%	94.6%
	Other	0.90 (0.45-1.81)	0.98 (0.37-2.63)	100.0%	100.0%
	Polytherapy	2.02 (1.49-2.75)	2.21 (1.48-3.30)	91.1%	85.0%
ADHD	No ASM	1 (Reference)	1 (Reference)	-	-
	Carbamazepine	1.07 (0.94-1.21)	0.98 (0.77-1.24)	94.6%	82.7%
	Gabapentin	1.22 (0.92-1.63)	1.30 (0.91-1.86)	81.5%	70.4%
	Lamotrigine	0.89 (0.78-1.01)	0.87 (0.75-1.02)	94.8%	93.5%
	Levetiracetam	1.05 (0.64-1.74)	1.12 (0.63-2.02)	90.9%	87.5%
	Phenytoin	1.16 (0.79-1.70)	2.72 (1.13-6.56)	96.3%	80.0%
	Pregabalin	0.67 (0.55-0.83)	0.76 (0.62-0.95)	96.8%	96.6%
	Topiramate	0.80 (0.49-1.30)	0.95 (0.58-1.55)	94.0%	94.0%
	Valproate	1.20 (1.02-1.40)	1.18 (0.91-1.53)	95.5%	88.8%
	Other	0.95 (0.71-1.25)	0.97 (0.66-1.44)	98.0%	96.1%
	Polytherapy	1.12 (0.92-1.36)	1.12 (0.86-1.46)	93.8%	90.0%

10. Line 150, guanfacine is also licensed for ADHD in both countries. If exposure to guanfacine was not used to define ADHD cases, please kindly justify.

Answer: Thank you for allowing us to clarify this matter. Guanfacine is licensed for treatment of ADHD symptoms in young people, but only as a second/third line option if stimulants and atomoxetine prove ineffective and/or insufficient. It is also used to treat high blood pressure and is used off-label to treat anxiety. We have reservations about including guanfacine in our outcome definition due to its second/third-line nature.

The benefit of using stimulants to identify ADHD is that they are (nearly) exclusively used for ADHD, are the first-line treatment (together with psychosocial support), and are the most common form of treatment for ADHD in young people in both countries. We have now briefly justified this in the revised manuscript (first paragraph of the section on Children’s neurodevelopmental conditions):

“To further identify cases of ADHD, we also incorporated prescription information on licensed ADHD medications (recorded using ATC codes at both data sources), including methylphenidate, dexamfetamine, lisdexamfetamine, and atomoxetine. We chose not to include guanfacine due to its second/third-line nature.”

11. Line 159, have the authors considered also including other medications such as lithium, ADHD medications, anxiolytics, antihypertensives, steroids, antibiotics and opioids as well? They can be potential confounders in this study.

Answer: Thank you for this important comment. We chose to adjust for medications that we assumed to be confounders, specifically those with a hypothesized causal effect on both anti-seizure medication exposure and neurodevelopmental conditions in children. While some of the drugs mentioned by the reviewer may influence the risk of neurodevelopmental conditions in children, it is difficult to envision them having a causal effect on the use of antiseizure medications. Therefore, we did not consider including these additional medications in our analysis; without an effect on the exposure, they would not lead to confounding.

We note, however, that we did control for many of the upstream indications of the drugs proposed by the reviewer (e.g., maternal ADHD, psychiatric conditions, pain conditions, etc.), so presumably we have accounted for these to some degree. Furthermore, the sibling analysis likely accounts for underlying maternal liability to indications justifying these and many other related medications.

As with any observational study, however, there is always a risk that we have not accounted for all confounders. We now acknowledge this as a limitation, also highlighting the possibility that we may not have accounted for all relevant co-prescriptions of therapeutics (third paragraph of the strengths and limitations):

“Nonetheless, like any safety analysis on this topic, our study is observational, and there is always a risk that we have not accounted for all relevant confounders or other co-prescriptions.”

12. Line 160, vomiting or antiemetic prescriptions during pregnancy were listed as covariates and in theory these could occurred after the occurrence of the exposure and thus could potentially mediated the results.

Answer: Thank you for bringing up this important point. Our rationale for adjusting for antiemetics was to indirectly adjust for emesis during pregnancy, which might result in lower-than-expected exposed dosages. Specifically, gravidas with persistent vomiting early in the trimester might not transfer the expected dose to the fetus, thereby distorting their exposure status.

We recognize that antiemetics could act as mediators by occurring after the exposure. Therefore, we have re-run the analysis without adjustment for antiemetic prescriptions and found that it does change the hazard ratio estimates from the primary analysis, but only beyond the second decimal.

We now comment on this briefly in the methods and results sections (see new subsections on sensitivity analyses).

13. Line 163, the supplementary table quoted does not describe the covariates.

Answer: Thank you for bringing this to our attention. This has now been corrected and referred to as the Supplementary Methods Covariates section.

14. Line 181, to account for potential exposure misclassification, have the authors considered limiting the exposed cohort to patients who have at least 2 prescriptions as one of the secondary analyses?

Answer: Thank you for this suggestion. We noted the use of this analysis in the US study and agree that it is an effective method to identify persistent users. We have now performed such analysis, finding that results are broadly comparable across ASMs and NDDs with the exception of Topiramate and ID where the effect estimate becomes larger in magnitude. We briefly comment on this in the revised methods and results (see new subsections on sensitivity analyses). However, we note that this analysis is largely comparable to studying ‘ASM continuers’, and we have previously shown that these individuals differ from those who can discontinue ASM use early in pregnancy (i.e., more epilepsy diagnoses and different covariate patterns among those who continue treatment throughout pregnancy; further details can be found in Madley-Dowd et al. 2023, ref #25).

For reviewer reference, below is an excerpt of the table summarizing the results of this analysis:

		HR (95% CI)		Weight for Sweden	
		Primary analysis	2 prescriptions for exposure definition	Primary analysis	2 prescriptions for exposure definition
ASD	No ASM	1 (Reference)	1 (Reference)	-	-
	Carbamazepine	1.25 (1.05-1.48)	1.35 (1.02-1.78)	88.0%	72.1%
	Gabapentin	1.15 (0.84-1.59)	1.15 (0.72-1.84)	56.0%	45.9%
	Lamotrigine	0.86 (0.72-1.01)	0.93 (0.76-1.14)	88.1%	83.0%
	Levetiracetam	0.82 (0.44-1.54)	1.17 (0.62-2.19)	80.3%	80.2%
	Phenytoin	0.99 (0.55-1.80)	1.25 (0.19-8.45)	90.3%	0.0%
	Pregabalin	0.71 (0.54-0.93)	0.84 (0.60-1.18)	87.1%	83.3%
	Topiramate	1.14 (0.68-1.93)	1.20 (0.60-2.40)	92.9%	87.6%
	Valproate	1.78 (1.48-2.14)	1.76 (1.30-2.37)	89.0%	74.6%
	Other	1.30 (0.93-1.84)	1.31 (0.80-2.14)	91.0%	93.4%
	Polytherapy	1.51 (1.22-1.88)	1.57 (1.23-2.00)	81.2%	76.5%
ID	No ASM	1 (Reference)	1 (Reference)	-	-
	Carbamazepine	1.30 (1.01-1.69)	1.50 (0.97-2.31)	94.0%	87.9%
	Gabapentin	1.02 (0.49-2.15)	0.73 (0.18-2.92)	100.0%	100.0%
	Lamotrigine	1.12 (0.86-1.46)	1.20 (0.87-1.67)	93.4%	90.0%
	Levetiracetam	0.66 (0.21-2.05)	0.98 (0.31-3.08)	100.0%	100.0%
	Phenytoin	0.67 (0.22-2.10)	2.13 (0.30-15.26)	100.0%	100.0%
	Pregabalin	0.86 (0.52-1.40)	0.71 (0.34-1.50)	100.0%	100.0%
	Topiramate	2.48 (1.23-4.98)	4.36 (2.06-9.23)	87.7%	100.0%
	Valproate	2.56 (1.97-3.32)	2.83 (1.85-4.33)	96.7%	91.3%

	Other	0.90 (0.45-1.81)	1.04 (0.39-2.81)	100.0%	100.0%
	Polytherapy	2.02 (1.49-2.75)	1.93 (1.32-2.81)	91.1%	86.4%
ADHD	No ASM	1 (Reference)	1 (Reference)	-	-
	Carbamazepine	1.07 (0.94-1.21)	1.02 (0.80-1.30)	94.6%	82.1%
	Gabapentin	1.22 (0.92-1.63)	0.83 (0.49-1.39)	81.5%	77.7%
	Lamotrigine	0.89 (0.78-1.01)	0.95 (0.80-1.11)	94.8%	92.5%
	Levetiracetam	1.05 (0.64-1.74)	1.23 (0.70-2.16)	90.9%	88.3%
	Phenytoin	1.16 (0.79-1.70)	1.71 (0.71-4.13)	96.3%	80.0%
	Pregabalin	0.67 (0.55-0.83)	0.75 (0.57-0.98)	96.8%	96.3%
	Topiramate	0.80 (0.49-1.30)	0.79 (0.40-1.59)	94.0%	100.0%
	Valproate	1.20 (1.02-1.40)	1.26 (0.96-1.65)	95.5%	89.6%
	Other	0.95 (0.71-1.25)	0.93 (0.62-1.41)	98.0%	100.0%
	Polytherapy	1.12 (0.92-1.36)	1.14 (0.91-1.43)	93.8%	91.5%

15. The difference between the exposed and the unexposed groups were clearly demonstrated in Table 1. Some of these differences may lead to residual confounding even after adjustment in the analysis and in the sibling analysis. Propensity score is generally applied not only to address confounding but also to demonstrate if the exposure groups are comparable. It would be helpful if the investigator could discuss the comparability between the groups.

Answer: Thank you for this important comment. We agree that Table 1 highlights differences between the exposed and unexposed groups, but in multivariable regression we have adjusted for factors that differ. In addition to that, we expect that these differences are smaller when compared to an active comparator (i.e., lamotrigine) – or children born to the same mother (sibling analysis).

However, it is important to recognize that, as with any observational study, there is an inherent risk of residual confounding (e.g., incorrectly measured confounders) or unobserved confounders (i.e., those for which we do not have data), despite using rigorous multivariable regression, active comparators and sibling analysis. Therefore, we have added a further discussion point on the differences between users and non-users, building on the sentence regarding co-prescriptions previously suggested by the reviewer (3rd paragraph of the strengths and limitations section of the discussion):

“Nonetheless, like any safety analysis on this topic, our study is observational, and there is always a risk that we have not accounted for all relevant confounders or other co-prescriptions. It may be important to recognize the differences between users and non-users of ASMs (Table 1) and consider these when interpreting our findings; even when comparing to lamotrigine users, or siblings to each other, important differences may persist.”

Authors may also consider adding mental health comorbidities at baseline in Table 1.

Answer: Thank you for this comment. Maternal psychiatric indications are presented in Table 1, and we believe this captures the relevant mental health comorbidities.

16. Typo in Table 1, calendar year of delivery, 2019-20 for the unexposed group in Sweden. It should be 245890.

Answer: Thank you for bringing this to our attention, we have now corrected this.

17. In table S10, it is noted that the limited number of patients were identified for some of the outcomes for individual antiseizure medications. It may worth addressing that in the limitation section under discussion.

Answer: Thank you for allowing us to expand on this matter. Table S10 includes outcome counts for individual antiseizure medications stratified by indications. While these counts are indeed limited, we are not overly concerned about this as we do not infer or conclude much from the indication-stratified data. Nevertheless, we agree with the reviewer's sentiment that further large-scale studies are always beneficial. We have now elaborated on this point in the discussion (final paragraph of strengths and limitations).

“Although our combined efforts across the UK and Sweden have allowed us to amass one of the largest and longest follow-ups of children diagnosed with neurodevelopmental conditions following ASM exposure in pregnancy, we emphasize that further data on ASM safety are warranted. This need becomes more pertinent as the landscape of ASM continues to evolve, given changing indication prevalences, costs, and known teratogenicity profiles. To enable such large-scale analysis and enhance generalizability, it will remain crucial to involve data from multiple countries across the globe.”

18. Line 225, the phrase “in addition” was repeated.

Answer: Thank you for bringing this to our attention, we have now corrected this (second paragraph of primary analyses section in the results).

19. Line 234-235, table should be S8 instead of S7.

Answer: Thank you for bringing this to our attention, we have now corrected this.

20. Line 250, as stated in the supplementary information, why is neuropathic pain not included as somatic indications in the Swedish cohort? There are discrepancies for the somatic indications between the UK and Swedish cohort. It will be much appreciated if the authors can explain such difference.

Answer: Thank you for allowing us to clarify this matter. As outlined in our response to comment #3, during the implementation of the common analytical protocol, we encountered a limitation in the Swedish data regarding the resolution of neuropathic pain ICD-codes. Specifically, the data were truncated by the registry holders/Swedish authorities to a broader category (ICD 10: M70-

M79.9) to safeguard individual privacy, given the rarity of these conditions in the pregnancy population. As we still wanted to control for these somatic indications, we instead expanded the definition in Sweden to more broadly defined chronic pain, migraine and diabetes complications (i.e., neuropathy). In practical terms, the difference between employing specific diagnostic codes and adopting a broader classification strategy likely has minimal impact, considering that we capture a broader range of conditions, and the relative rarity of these conditions within the pregnancy population.

We have provided additional clarification on this matter in the supplementary methods (section titled Covariates – subsection Antiseizure medication (ASM) indication definitions).

21. Line 286-287, according to Supplementary Table S13, some of the HRs of Carbamazepine associated with autism and ID are not significantly larger than 1 when analysis was stratified by different indications. This contradicts to this statement.

Answer: Thank you for allowing us to expand on this matter. The relevant section is meant to convey that even after adjusting for indications using multivariable regression, the findings persist — some prior studies have suggested that the findings are explained by confounding by indication.

Furthermore, we note that there is limited heterogeneity between our main analysis and the stratified analysis (see Table S7), even if even if they have more uncertainty (which is to be expected due to the decreased sample size) and therefore are not statistically significant. However, as outlined above regarding Table S10, we do not wish to infer too much from the indication-stratified analysis because we have previously shown that such analyses are generally more biased and do not provide a methodologically sound way to adjust for indication in observational studies (see Ahlqvist et al. 2023, ref #35). Briefly, such analysis tends to exaggerate existing biases in observational studies, both by bias amplification and by introducing random noise, without providing a ‘better’ way to account for indication. This is why we caution against its interpretation in the original submission (lines 258-261).

Nevertheless, we have now revised the relevant sentence to read:

“Our study provides evidence suggesting that carbamazepine is associated with an increased likelihood of autism and ID after accounting for indication through multivariable regression, albeit there is still some uncertainty in the indication stratified data”

22. Line 296, the main analysis also shows that there is a higher risk for autism.

Answer: Thank you for bringing this to our attention. The relevant line (296) refers to the consistency across the main and sibling analyses, where the findings for valproate were consistent

for ADHD and intellectual disability but less so for autism. However, considering the add-on sentence, we agree that the text could be revised to read (third paragraph of the comparison with previous literature section):

“Our findings support this for ID, ADHD, and autism, though our sibling analyses which better accounted for potential unobserved genetic and environmental showed substantially attenuated associations between valproate and autism which potentially implicates unmeasured confounding.”

23. Line 297-298, the HR for ID was also attenuated and confidence intervals cover 1, should the authors consider mentioning it in this statement as well?

Answer: Thank you for pointing this out. We were not specifically focusing on whether a particular estimate was 'statistically significant' in the sibling analysis, but rather on qualitatively assessing the overall consistency of the results. However, as we agree that this could be pointed out, we have revised the section to read (also third paragraph of the comparison with previous literature section):

“Our findings support this for ID, ADHD, and autism, though our sibling analyses which better accounted for potential unobserved genetic and environmental showed substantially attenuated associations between valproate and autism which potentially implicates unmeasured confounding. These analyses need further replication as they are lower powered than traditional analyses. For example, the risk of intellectual disability in the sibling analysis excluded the null in Sweden (HR: 2.23; 95% CI: 1.01-4.92), but wide uncertainties in the UK pulled down the pooled estimate, resulting in confidence intervals that did not exclude the null (HR: 1.54; 95% CI: 0.76-3.14).”

Reviewer #3 (Remarks to the Author):

This is a substantial study of UK and Swedish data investigating exposures to ASM and outcomes of autism, ADHD and intellectual disability. The study includes full populations from two electronic healthcare databases of women with pregnancies ending in live births that include the years 1995-2018 (or 2020 for Sweden). Mothers needed 1 year of data before the start of their pregnancy and to be followed up until the end of their pregnancy. Analyses of ASM exposed versus unexposed in the full population and secondary analyses of siblings and using lamotrigine as an active comparator are included. An extensive set of results are presented in the supplementary information supplied by the authors.

Answer: Thank you for taking the time to review our work. We believe your thoughtful comments have helped us improve the reporting of our study.

It is appreciated that word counts can limit the inclusion of all details that might be needed to fully explain the work presented and the supplementary information can enable more to be included, however some further clarification of the following would be helpful in the main paper:

Answer: Thank you for your comment. We have taken your feedback, along with input from other reviewers, into careful consideration. As a result, we have expanded certain sections of the main article and integrated content from the supplementary material into the main text.

Follow up: Please comment on whether there is any impact of including children born in 2020 but only followed up for 1 year (Sweden data) given that it is unlikely that the outcomes of interest could be detected by one year of age (apart from very profound developmental delay). Would a minimum follow up time for the children born following the included pregnancies give more potential for diagnoses of the outcomes of interest e.g. child development checks at pre-school ages?

Answer: Thank you for your insightful comment. We did deliberate on the inclusion of the Swedish 2020 cohort during the protocol development phase and decided to incorporate them, considering their relatively minor impact.

However, we acknowledge the importance of sufficient follow-up when evaluating the antiseizure medication impact on children's neurodevelopmental conditions. In fact, we believe that our study stands out in this regard compared to previous research, as indicated by the median age of follow-up in our cohort (for example, the US study had a median follow-up of 2 years, while we had 7.5 years in the UK and 12.5 years in Sweden). Conducting time-to-event analysis in a cohort with a longer median age of follow-up allows us to capture potential differences in early diagnoses, but also ensures that we capture ages where diagnoses are more frequently identified (i.e., preschool ages where child development checks are often conducted).

Sharing the reviewers' interest in cohorts where these conditions are expected to be more frequently diagnosed, we have now also included supplementary analyses where we replicate our main analysis in cohorts born before 1st of Jan 2018 and 1st of Jan 2017 in Sweden and UK respectively, ensuring a minimum follow-up of 4 years in both Sweden and the UK (i.e., pre-school age). The results of these sensitivity analyses show that restricting the cohort to ensure a minimum follow-up of 4 years does not change the conclusions of our primary analyses.

We briefly comment on these results in the revised methods and results sections (new subsections on sensitivity analyses) and include a table of them in the supplementary results (Table S13).

Methods: “We conducted the following sensitivity analyses: *1) we ensured a minimum follow-up of 4 years, replicating our primary analysis in cohorts born before 1st of Jan 2018 and 1st of Jan 2017 in Sweden and the UK respectively; ...*”

Results: “Results of sensitivity analyses 1-3 are presented in supplementary Table S13. *Restricting the cohorts to ensure a minimum follow-up of 4 years did not change the conclusions of our primary analyses....*”

For reviewer reference, below is an excerpt of the table summarizing the results of this analysis:

		HR (95% CI)		Weight for Sweden	
		Primary analysis	Restricted cohort analysis	Primary analysis	Restricted cohort analysis
ASD	No ASM	1 (Reference)	1 (Reference)	-	-
	Carbamazepine	1.25 (1.05-1.48)	1.17 (0.98-1.38)	88.0%	88.1%
	Gabapentin	1.15 (0.84-1.59)	1.23 (0.89-1.69)	56.0%	56.0%
	Lamotrigine	0.86 (0.72-1.01)	0.91 (0.77-1.08)	88.1%	88.0%
	Levetiracetam	0.82 (0.44-1.54)	1.02 (0.54-1.90)	80.3%	80.4%
	Phenytoin	0.99 (0.55-1.80)	0.84 (0.47-1.52)	90.3%	90.3%
	Pregabalin	0.71 (0.54-0.93)	0.70 (0.53-0.92)	87.1%	92.1%
	Topiramate	1.14 (0.68-1.93)	1.04 (0.60-1.80)	92.9%	92.4%
	Valproate	1.78 (1.48-2.14)	1.67 (1.39-2.00)	89.0%	89.1%
	Other	1.30 (0.93-1.84)	1.20 (0.85-1.69)	91.0%	91.0%
	Polytherapy	1.51 (1.22-1.88)	1.50 (1.20-1.86)	81.2%	81.7%
ID	No ASM	1 (Reference)	1 (Reference)	-	-
	Carbamazepine	1.30 (1.01-1.69)	1.26 (0.97-1.63)	94.0%	94.0%
	Gabapentin	1.02 (0.49-2.15)	0.77 (0.32-1.85)	100.0%	100.0%
	Lamotrigine	1.12 (0.86-1.46)	1.18 (0.91-1.53)	93.4%	93.4%
	Levetiracetam	0.66 (0.21-2.05)	0.77 (0.25-2.41)	100.0%	100.0%
	Phenytoin	0.67 (0.22-2.10)	0.62 (0.20-1.93)	100.0%	100.0%
	Pregabalin	0.86 (0.52-1.40)	0.89 (0.55-1.47)	100.0%	100.0%
	Topiramate	2.48 (1.23-4.98)	2.55 (1.27-5.12)	87.7%	87.7%
	Valproate	2.56 (1.97-3.32)	2.42 (1.86-3.14)	96.7%	96.7%
	Other	0.90 (0.45-1.81)	0.88 (0.44-1.77)	100.0%	100.0%
	Polytherapy	2.02 (1.49-2.75)	2.07 (1.52-2.81)	91.1%	91.1%
ADHD	No ASM	1 (Reference)	1 (Reference)	-	-
	Carbamazepine	1.07 (0.94-1.21)	0.99 (0.87-1.12)	94.6%	94.7%
	Gabapentin	1.22 (0.92-1.63)	1.24 (0.94-1.65)	81.5%	81.5%
	Lamotrigine	0.89 (0.78-1.01)	0.95 (0.83-1.08)	94.8%	94.8%
	Levetiracetam	1.05 (0.64-1.74)	1.23 (0.74-2.03)	90.9%	90.8%
	Phenytoin	1.16 (0.79-1.70)	0.96 (0.66-1.41)	96.3%	96.3%
	Pregabalin	0.67 (0.55-0.83)	0.70 (0.57-0.86)	96.8%	97.8%
	Topiramate	0.80 (0.49-1.30)	0.82 (0.50-1.33)	94.0%	94.0%
	Valproate	1.20 (1.02-1.40)	1.11 (0.95-1.29)	95.5%	95.5%
	Other	0.95 (0.71-1.25)	0.87 (0.66-1.16)	98.0%	98.0%
	Polytherapy	1.12 (0.92-1.36)	1.12 (0.92-1.36)	93.8%	93.8%

Exposure: Please could you define how exposure is classified during the pregnancy. Is this by receipt of any ASM at any time or is this broken down by trimester or month of pregnancy? Are

changes in exposures during pregnancy accounted for in the analyses or did women use the same medication throughout? How is polytherapy defined?

Answer: Thank you for your valuable comment. In response to your suggestion, along with feedback from other reviewers, we have expanded the description of the exposure in the main article by moving relevant details from the supplementary material.

In the original submission, the exposure was defined as any antiseizure medication used at any time during pregnancy. We define it as such because it aligns with existing literature, and because analyzing specific drugs and specific neurodevelopmental outcomes at specific timepoints in pregnancy is not as well-powered. However, in response to your and other reviewers' interest, we conducted additional analyses where we examine first-trimester use, given its relevance to fetal development. These analyses are now included in the revised methods and results sections (see sensitivity analysis subsections) and in the revised Table S13 (an excerpt of which is found below).

Methods: “We conducted the following sensitivity analyses... 2) we examined exposure in the first trimester, given its relevance to fetal development.”

Results: “When exploring exposure during the first trimester, all ASM and NDD combinations were comparable to the primary analyses except for phenytoin, which became associated with ADHD with large uncertainty.”

Briefly, we find that results for all ASM and NDD combinations are comparable to the primary analyses. One exception was phenytoin, which became associated with ADHD, albeit with large uncertainty. Nonetheless, phenytoin is a well-known teratogen and typically already contraindicated during pregnancy.

		HR (95% CI)		Weight for Sweden	
		Primary analysis	Trimester 1 analysis	Primary analysis	Trimester 1 analysis
ASD	No ASM	1 (Reference)	1 (Reference)	-	-
	Carbamazepine	1.25 (1.05-1.48)	1.35 (1.03-1.77)	88.0%	72.1%
	Gabapentin	1.15 (0.84-1.59)	1.19 (0.82-1.71)	56.0%	45.6%
	Lamotrigine	0.86 (0.72-1.01)	0.92 (0.76-1.11)	88.1%	85.7%
	Levetiracetam	0.82 (0.44-1.54)	1.27 (0.70-2.32)	80.3%	82.0%
	Phenytoin	0.99 (0.55-1.80)	1.19 (0.18-8.05)	90.3%	0.0%
	Pregabalin	0.71 (0.54-0.93)	0.79 (0.60-1.05)	87.1%	86.3%
	Topiramate	1.14 (0.68-1.93)	1.33 (0.79-2.26)	92.9%	92.9%
	Valproate	1.78 (1.48-2.14)	1.61 (1.20-2.15)	89.0%	71.2%
	Other	1.30 (0.93-1.84)	1.34 (0.84-2.13)	91.0%	83.5%
	Polytherapy	1.51 (1.22-1.88)	1.52 (1.16-2.00)	81.2%	73.1%
ID	No ASM	1 (Reference)	1 (Reference)	-	-
	Carbamazepine	1.30 (1.01-1.69)	1.36 (0.87-2.12)	94.0%	87.0%
	Gabapentin	1.02 (0.49-2.15)	1.11 (0.46-2.66)	100.0%	100.0%

	Lamotrigine	1.12 (0.86-1.46)	1.12 (0.82-1.54)	93.4%	90.7%
	Levetiracetam	0.66 (0.21-2.05)	1.26 (0.46-3.40)	100.0%	100.0%
	Phenytoin	0.67 (0.22-2.10)	2.77 (0.39-19.84)	100.0%	100.0%
	Pregabalin	0.86 (0.52-1.40)	0.92 (0.54-1.57)	100.0%	100.0%
	Topiramate	2.48 (1.23-4.98)	2.47 (1.17-5.21)	87.7%	86.0%
	Valproate	2.56 (1.97-3.32)	2.06 (1.29-3.29)	96.7%	94.6%
	Other	0.90 (0.45-1.81)	0.98 (0.37-2.63)	100.0%	100.0%
	Polytherapy	2.02 (1.49-2.75)	2.21 (1.48-3.30)	91.1%	85.0%
ADHD	No ASM	1 (Reference)	1 (Reference)	-	-
	Carbamazepine	1.07 (0.94-1.21)	0.98 (0.77-1.24)	94.6%	82.7%
	Gabapentin	1.22 (0.92-1.63)	1.30 (0.91-1.86)	81.5%	70.4%
	Lamotrigine	0.89 (0.78-1.01)	0.87 (0.75-1.02)	94.8%	93.5%
	Levetiracetam	1.05 (0.64-1.74)	1.12 (0.63-2.02)	90.9%	87.5%
	Phenytoin	1.16 (0.79-1.70)	2.72 (1.13-6.56)	96.3%	80.0%
	Pregabalin	0.67 (0.55-0.83)	0.76 (0.62-0.95)	96.8%	96.6%
	Topiramate	0.80 (0.49-1.30)	0.95 (0.58-1.55)	94.0%	94.0%
	Valproate	1.20 (1.02-1.40)	1.18 (0.91-1.53)	95.5%	88.8%
	Other	0.95 (0.71-1.25)	0.97 (0.66-1.44)	98.0%	96.1%
	Polytherapy	1.12 (0.92-1.36)	1.12 (0.86-1.46)	93.8%	90.0%

Our main analyses disregard medication discontinuation late in pregnancy (i.e., they are invariant to changes in exposure status during pregnancy, since we rely on ‘any exposure in pregnancy’) but do account for therapeutic switching by defining polytherapy as the prescription or dispensation of at least two different antiseizure medications during pregnancy.

However, in response to another reviewer’s suggestion, we have conducted a sensitivity analysis where we define exposure by dispensation/prescription on at least two separate occasions during pregnancy (i.e., continued users). Using the same definition of polytherapy as in the main analysis, this secondary analysis essentially ensures that a woman uses the same therapeutic throughout a large part of the pregnancy. These analyses are now included in the revised methods and results sections (sensitivity analyses subsections), and an in Table S13, an excerpt of which is provided below for the convenience of the reviewer.

Methods: “We conducted the following sensitivity analyses: ... 3) we defined exposure by dispensation/prescription on at least two separate occasions during pregnancy (i.e., continued users), using the same definition of polytherapy as in the main analysis. This analysis makes it more likely that a woman uses the same therapeutic throughout a large part of the pregnancy.”

Results: “When modifying the exposure definition to require two prescriptions/dispensations, the results were comparable with the primary analyses for all ASM-outcome combinations, except topiramate and ID where the estimate was increased under the new definition.”

Briefly, we find that results are comparable with the primary analyses for all ASM and outcome combinations, except Topiramate and ID where the estimate was increased under the new definition. This provides further support for our conclusion that pregnancy use of topiramate is associated with an increased risk of neurodevelopmental conditions in offspring.

		HR (95% CI)		Weight for Sweden	
		Primary analysis	2 prescriptions for exposure definition	Primary analysis	2 prescriptions for exposure definition
ASD	No ASM	1 (Reference)	1 (Reference)	-	-
	Carbamazepine	1.25 (1.05-1.48)	1.35 (1.02-1.78)	88.0%	72.1%
	Gabapentin	1.15 (0.84-1.59)	1.15 (0.72-1.84)	56.0%	45.9%
	Lamotrigine	0.86 (0.72-1.01)	0.93 (0.76-1.14)	88.1%	83.0%
	Levetiracetam	0.82 (0.44-1.54)	1.17 (0.62-2.19)	80.3%	80.2%
	Phenytoin	0.99 (0.55-1.80)	1.25 (0.19-8.45)	90.3%	0.0%
	Pregabalin	0.71 (0.54-0.93)	0.84 (0.60-1.18)	87.1%	83.3%
	Topiramate	1.14 (0.68-1.93)	1.20 (0.60-2.40)	92.9%	87.6%
	Valproate	1.78 (1.48-2.14)	1.76 (1.30-2.37)	89.0%	74.6%
	Other	1.30 (0.93-1.84)	1.31 (0.80-2.14)	91.0%	93.4%
	Polytherapy	1.51 (1.22-1.88)	1.57 (1.23-2.00)	81.2%	76.5%
ID	No ASM	1 (Reference)	1 (Reference)	-	-
	Carbamazepine	1.30 (1.01-1.69)	1.50 (0.97-2.31)	94.0%	87.9%
	Gabapentin	1.02 (0.49-2.15)	0.73 (0.18-2.92)	100.0%	100.0%
	Lamotrigine	1.12 (0.86-1.46)	1.20 (0.87-1.67)	93.4%	90.0%
	Levetiracetam	0.66 (0.21-2.05)	0.98 (0.31-3.08)	100.0%	100.0%
	Phenytoin	0.67 (0.22-2.10)	2.13 (0.30-15.26)	100.0%	100.0%
	Pregabalin	0.86 (0.52-1.40)	0.71 (0.34-1.50)	100.0%	100.0%
	Topiramate	2.48 (1.23-4.98)	4.36 (2.06-9.23)	87.7%	100.0%
	Valproate	2.56 (1.97-3.32)	2.83 (1.85-4.33)	96.7%	91.3%
	Other	0.90 (0.45-1.81)	1.04 (0.39-2.81)	100.0%	100.0%
	Polytherapy	2.02 (1.49-2.75)	1.93 (1.32-2.81)	91.1%	86.4%
ADHD	No ASM	1 (Reference)	1 (Reference)	-	-
	Carbamazepine	1.07 (0.94-1.21)	1.02 (0.80-1.30)	94.6%	82.1%
	Gabapentin	1.22 (0.92-1.63)	0.83 (0.49-1.39)	81.5%	77.7%
	Lamotrigine	0.89 (0.78-1.01)	0.95 (0.80-1.11)	94.8%	92.5%
	Levetiracetam	1.05 (0.64-1.74)	1.23 (0.70-2.16)	90.9%	88.3%
	Phenytoin	1.16 (0.79-1.70)	1.71 (0.71-4.13)	96.3%	80.0%
	Pregabalin	0.67 (0.55-0.83)	0.75 (0.57-0.98)	96.8%	96.3%
	Topiramate	0.80 (0.49-1.30)	0.79 (0.40-1.59)	94.0%	100.0%
	Valproate	1.20 (1.02-1.40)	1.26 (0.96-1.65)	95.5%	89.6%
	Other	0.95 (0.71-1.25)	0.93 (0.62-1.41)	98.0%	100.0%
	Polytherapy	1.12 (0.92-1.36)	1.14 (0.91-1.43)	93.8%	91.5%

Outcome: Please define intellectual disability further - does this include learning difficulty and/or learning disability and /or IQ scores?

Answer: Thank you for allowing us to clarify this. Intellectual disability, along with the other outcomes, was identified based on recorded diagnoses in healthcare registries. This diagnosis encompasses deficits in both intellectual and adaptive functioning across various domains, including conceptual, social, and practical skills, typically presenting during the developmental period. Historically, intellectual disability was often defined by an intelligence quotient lower than 70, but recent changes have shifted from this being a strict criterion to intelligence testing being one tool in a more comprehensive assessment.

In the revised manuscript, we have expanded on the definition on all three outcomes (second paragraph of the Children’s neurodevelopmental conditions):

“Autism is characterized by impairments in social communication and interaction, along with restricted and repetitive behaviors or interests. ADHD involves symptoms in two primary domains: hyperactivity/impulsivity and inattention, with varying extents and presentations. Intellectual disability entails deficits in both intellectual and adaptive functioning, affecting conceptual, social, and practical skills, and typically manifests during the developmental period. Historically, intellectual disability was defined by an intelligence quotient below 70. However, diagnostic criteria have evolved to encompass a more comprehensive assessment, rather than relying solely on intelligence scores. Nonetheless, autism, ADHD and intellectual disability frequently co-occurs.”

Please also state whether the prevalence of the outcomes achieved is as expected in the two populations and if these are not as expected then suggest why this is the case. The supplementary file indicates that the numbers with ID in the UK are very low (S2) so more explanation about whether cases are missed or not in the presentation of these results is needed.

Answer: Thank you for allowing us to elaborate on this point. It may be important to acknowledge that that the registries we rely on do not cover all routes to a diagnosis. In Sweden, we capture inpatient and specialized outpatient healthcare, while in the UK, we capture general practice and hospital care (for the 80% with HES coverage). However, the numbers we present align with prior studies conducted in Sweden (see our prior work: PMID: 38592388) and using the CPRD in the UK (e.g., Russel et al. 2022 PMID: 34414570).

As the reviewer alludes to, and as we noted in the original submission (e.g., line 310-312), there are differences in the prevalence of these diagnoses between Sweden and the UK. We believe this presents a valuable opportunity to compare and triangulate findings across different healthcare settings/data sources. That is, despite these differences, the associations we observe are largely consistent across both countries, which may provide some reassurance regarding the validity of our conclusions.

Nonetheless, we acknowledge that intellectual disability is diagnosed more frequently in Sweden than in the UK (even when considering non-CPRD based studies, see PMID: 2633045). We have now further expanded on this in the limitation section (first paragraph of the section), also considering the subsequent comment regarding maternal neurodevelopmental conditions:

“Differences in covariates across data sources were observed ... and are likely due to the differing nature of the underlying data sources (such as the capture of primary or specialist outpatient care). It may be important to note that we observed differences in outcome frequencies, with Sweden having nearly twice the diagnostic prevalence of neurodevelopmental conditions in children, and even more so in mothers, albeit this is consistent with previous studies in the UK (PMID: 34414570) and Sweden (PMID: 38592388). Despite these differences, we found comparable effect estimates for most monotherapies (see data source-specific estimates and tests of heterogeneity in the supplementary results).”

Results: it was noted that diagnoses of mothers with NDD is much higher in Sweden than in the UK. Why is this? Are diagnostic criteria or methods of case identification different?

Answer: Thank you for allowing us to clarify this. We believe this discrepancy likely arises from the same factors influencing child outcomes, as outlined in our response to the prior comment. There are differences in the underlying data contributing to our data sources (CPRD relying on general practice/hospital data and Sweden on inpatient/specialized outpatient data), and there are also variations in diagnostic preference and tradition between the settings.

As discussed in the prior comment, we view these differences as an opportunity to triangulate the associations across settings with clear distinctions, thereby enhancing the validity of the associations identified in both settings. In other words, by combining data sources with differences, it allows us to strengthen the robustness of our findings by accounting for variations in healthcare practices and diagnostic patterns between Sweden and the UK.

Given the thematic similarity between this comment and the previous one, we have adapted the revisions made in the updated manuscript to address both maternal and child neurodevelopmental conditions (see first paragraph of the strengths and limitations):

“Differences in covariates across data sources were observed ... and are likely due to the differing nature of the underlying data sources (such as the capture of primary or specialist outpatient care). It may be important to note that we observed differences in outcome frequencies, with Sweden having nearly twice the diagnostic prevalence of neurodevelopmental conditions in children, and even more so in mothers, albeit this is consistent with previous studies in the UK (Russell et al., 2022, PMID: 34414570) and Sweden (PMID: 38592388). Despite these differences, we found comparable effect estimates for most

monotherapies (see data source-specific estimates and tests of heterogeneity in the supplementary results).”

Supplementary file: Please could you add footnotes to state what the counts of outcomes represent - whether this is at 4, 8 or 12 years or for the full study.

Answer: We assume this comment pertains tables S2, S10 in the supplementary results. In response, we have added a footnote (“Number with outcome at the end of follow up”) to these tables to ensure clarity regarding what each count represents.

Is there scope to include aspects of S10 and S13 in the main paper since there is little published about some of these exposures and these results may prove helpful to those contemplating pregnancy and their healthcare workers. However some editing would be needed to highlight the channelling of some medications towards one set of maternal indications over others e.g. pregabalin or gabapentin are unlikely to be prescribed for epilepsy.

Answer: Thank you for giving us the opportunity to incorporate further results from the supplementary material into the manuscript. We have thoroughly reviewed the supplementary results and decided to include a revised version of Table S13 from the originally submitted manuscript in the main article (now Table 2), where we present the combined findings stratified across indications. As we detailed in the original manuscript, however, we have reservations about the biases inherent in indication-stratified analyses.

Extended figure 1: Could the epilepsy only data for this study in the figure at the relevant points to give this comparison?

Answer: Thank you for your suggestion. To ensure consistency and cater to broader interest in different types of indications, we have included both epilepsy, psychiatric, and somatic stratifications in the revised figure.

Overall, this is an extensive piece of work that is well described and will be of significance in the field. The challenge in presenting this work though is that there is a lot of detail to be described and data that is needed to fully understand what has been found to enable this to be put into context with others' work.

Answer: Thank you for your meticulous review and endorsement of our work. In response to your and other reviewers' suggestions, we have diligently worked to enhance the reporting of our study by incorporating various aspects from the supplementary material into the main article. We acknowledge that large-scale efforts such as ours require extensive details and data, which is why we chose to include a comprehensive supplementary methods and results section. Our intention was to support ongoing regulatory efforts with this detailed information.

Nevertheless, we have summarized the key findings of our paper in the discussion, conclusion, and abstract sections, ensuring that readers looking for a quick take-home message can easily access these summaries. Furthermore, to assist readers in contextualizing our work within the broader field, we have also summarized recent large-scale studies on this topic and quantitatively compared the findings related to carbamazepine.

I would prefer to see extra supplementary tables from the results given in the main paper and the extended data table comparison (page 24) (while interesting to review) to be included in the supplementary file.

Answer: Thank you for this feedback. In light of your comments, and those of other reviewers, we have moved data from the supplement into the main article (see rebuttal above regarding Table S13). We believe that the extended data table comparison and extended data figure help readers place our work in a broader context; we refer to it both in the introduction and the discussion (e.g., summarizing recent studies, and comparing to other studies of carbamazepine). In light of this, and the display item limits of Nature Communications we chose to keep this in the main article.

REVIEWER COMMENTS

Reviewer #1 & #2 (Remarks to the Author):

We would like to thank the research team for their careful attention to our comments. The updated version addresses some of the points raised in the initial review. However, we have a few additional suggestions that will further enhance the robustness of the study.

Answer: We are pleased that the reviewers appreciated our revisions, and we are grateful for their attention to detail. We have carefully addressed and revised their remaining comments, and we agree that the rigor of our study has been strengthened thanks to the reviewers' valuable input.

1. In response to the first comment, we acknowledge that the authors have revised the sentence in Lines 83-87. While it is true that the current study may have an advantage in follow-up time and sample size for Topiramate, the overall cohort, which includes various ASMs, has a smaller sample size and shorter follow-up time compared to the referenced Nordic study.

Answer: Thank you for allowing us to clarify. The referenced Nordic study indeed had a larger total sample size (N=4,494,926); albeit, they do not appear to have adjusted for epilepsy in their overall analysis, so the most comparable hazard ratios likely come from a sample of N=37,804 children born to individuals with epilepsy.

The reviewer acknowledges that our sample size and follow-up of Topiramate are larger than the prior Nordic study. In contrast to the reviewer's remark, we wish to clarify that this is also true across all other medications examined in both our study and the Nordic study. That is, our combined CPRD/Sweden sample has a *longer* follow-up period—though it is often comparable between the Nordic study and CPRD specifically (see below). Nevertheless, lines 83-87 are predominantly to note the US study's short follow-up (i.e., a median of 2 years), which we feel has received insufficient attention in the discussion of its weight to the total evidence.

For these reasons, we choose to retain the sentence as is.

	Bjørk et al.	Our study		
	Total median follow-up time (IQR)	Pooled median follow-up time	CPRD median follow-up time (IQR)	Sweden median follow-up time (IQR)
Total cohort	8.0 (4.0-12.1)	11.7	7.5 (4.1-11.6)	12.5 (6.5-19.3)
Lamotrigine	5.1 (2.4-8.5)	6.4	6.2 (3.5-9.9)	6.5 (3.6-10.7)
Carbamazepine	10.5 (6.4-16.6)	14.5	9.2 (5.3-12.5)	15.8 (9.5-21.1)
Valproate	9.9 (6.0-14.3)	13.4	9.3 (5.5-12.8)	14.9 (9.8-19.7)
Pregabalin	5.2 (2.7-7.5)	7.0	4.9 (3.1-7.3)	7.7 (4.6-10.7)
Gabapentin	3.5 (1.7-6.7)	5.1	4.9 (3.1-7.5)	5.2 (3.1-9.3)
Levetiracetam	3.6 (1.8-6.5)	4.4	6.2 (3.5-9.9)	6.5 (3.6-10.6)

Topiramate	5.7 (2.9-9.1)	7.2	5.4 (3.6-8.4)	8.2 (4.2-12.1)
------------	---------------	-----	---------------	----------------

2. In response to the second comment, in the Supplementary Methods (pages 9-13), it is not only the definition of neuropathic pain that differs. It would be clearer for readers if it were explicitly noted that the definitions of covariates and comorbidities also vary between the two databases, following consultation with local experts. Additionally, it would be helpful to explain to readers why these differences exist and whether they would affect the interpretation of the results.

Answer: Thank you for highlighting the need to clarify this to our readers. We have now made the following edits to the methods section on covariates and the strengths and limitations section.

Edit to the section on covariates (Lines 182-195, page 6):

“... Detailed definitions of all covariates, **which differed between countries due to the availability of data and coding systems employed by each data source and following consultation with local clinical experts,** can be found in the Supplementary Methods Covariates section. ...”

Edit to first paragraph of the limitations section (starting on line 356 of page 13):

“Differences in covariates across data sources were observed (e.g., the UK's lower percentage of mothers with no identified indication, 3.4% vs 11.0% among ASM users; the higher percentage of mothers with more than ten healthcare visits in the UK, 59.0% vs 7.2% among ASM users; and Sweden's higher percentage of mothers with neurodevelopmental conditions among ASM users, 17.5% vs 1.41%), and are likely due to the differing nature of the underlying data sources (such as the capture of primary or specialist outpatient care) **and resulting definitions used (see Covariates section of the Supplementary Methods).** It may be important to note that we observed differences in outcome frequencies, with Sweden having nearly twice the diagnostic prevalence of neurodevelopmental conditions in children, and even more so in mothers, albeit this is consistent with previous studies in the UK 38 and Sweden 39. Despite these differences, we found comparable effect estimates for most monotherapies (see data source-specific estimates and tests of heterogeneity in the Supplementary Results).”

3. In response to comment 5, Lines 122 and 299 still do not specify the exact parts or sections of the supplementary materials. For example, in Lines 298-299, the statement "Drug-specific estimates are reported in Table 2 and the Supplementary Results" should specify which particular table in the Supplementary Results. Similarly, in Lines 121-122, where it states, "Further

details on specific data sources at the county level can be found in the Supplementary Methods," please indicate the specific section of the Supplementary Methods.

Answer: Thank you for your attention to detail—we overlooked these instances in the previous revision. This has now been corrected. We have also carefully reviewed the entire manuscript for any other opportunities to more clearly reference specific sections of the supplementary material.

4. In response to comments 6 and 9, it is notable that valproate no longer shows a significantly increased risk of ADHD after restricting follow-up time to at least 4 years or when focusing on first-trimester exposure. The authors may want to highlight this finding to suggest that interpretation should be approached with caution. Since ADHD typically requires around 5 years for diagnosis, the shorter follow-up time in this study should also be addressed in the limitations section.

Answer: Thank you for bringing this to our attention. Upon re-reviewing the sensitivity analysis, we, too, were surprised by the lower association in the 4-year exclusion analysis, particularly since excluding the youngest participants was not expected to affect ADHD, a condition typically diagnosed later in childhood. This prompted a thorough internal and external review of all code scripts across both the UK and Sweden cohorts.

During this process, we identified a minor coding issue in the Swedish scripts. Specifically, when transitioning from the stpm3 to stpm2 package during revisions aimed at optimizing computation time, we inadvertently overlooked a key difference: stpm2 does not automatically apply certain corrections for omitted or difficult-to-estimate parameters, because it was developed back before such things were automated. Upon recognizing this, we re-estimated the affected results. While this led to subtle changes in some secondary estimates and sensitivity analyses, we confirm that none of our primary analyses, including sibling comparisons or the active comparator analysis, were impacted. Nevertheless, we apologize for this oversight.

In the corrected sensitivity analysis of valproate and ADHD with a minimum of 4 years follow-up the hazard ratio is statistically significant, as per expectation – the sensitivity analysis did not change the ADHD findings once the modelling issue was resolved (HR: 1.20; 95% CI 1.02-1.40, as compared to HR 1.20; 95% CI: 1.02-1.40 in the primary analysis). Nevertheless, the sensitivity analysis of the trimester-1 exposure and Dx2 exposure is not statistically significant (95% CI: 0.92-1.54 & 0.97-1.65, respectively). Although one might note that none of these sensitivity analyses are statistically significantly different from the primary estimates, we are inclined to agree with the reviewer that it is beneficial to be precise and consistent with our reporting and caveat that there is some uncertainty associated with the sensitivity analysis. We have therefore revised the results section line 303-304, page 10 to read:

“Results of sensitivity analyses 1-3 are presented in supplementary Table S13. **Briefly, the findings were largely consistent with the primary analysis, albeit some estimates were associated with more uncertainty. That is,** restricting the cohorts to ensure a minimum follow-up of 4 years did not change the conclusions of our primary analyses. When exploring exposure during the first trimester, all ASM and NDD combinations were comparable to the primary analyses except for phenytoin, which became associated with ADHD with large uncertainty. When modifying the exposure definition to require two prescriptions/dispensations, the results were comparable with the primary analyses for all ASM-outcome combinations, except topiramate and intellectual disability where the estimate was increased under the new definition. Repeating primary analyses excluding the covariate for vomiting or antiemetic prescriptions from the adjustment set only changed hazard ratio estimates beyond the second decimal (see Supplementary Results Table S14).

Finally, we are somewhat uncertain what the reviewer refers to by ‘shorter follow-up’, as we have the longest follow-up of all existing published studies that we are aware of (e.g., see Reviewer 1 Comment 1). Nevertheless, as added in prior revisions, we agree that further studies with even longer and larger samples are welcomed:

“..., we emphasize that further data on ASM safety are warranted. This need becomes more pertinent as the landscape of ASM continues to evolve, given changing indication prevalences, costs, and known teratogenicity profiles. To enable such large-scale analysis and enhance generalizability, it will remain crucial to involve data from multiple countries across the globe.”

5. In response to comment 11, the authors stated that “While some of the drugs mentioned by the reviewer may influence the risk of neurodevelopmental conditions in children, it is difficult to envision them having a causal effect on the use of antiseizure medications. Therefore, we did not consider including these additional medications in our analysis; without an effect on the exposure, they would not lead to confounding.” Please kindly share references to justify such claim.

If the investigator "believe" it is difficult to envision, they have to provide evidence to support such believe. The relationship between a potential confounding factor and the exposure can be simply a correlation, not necessarily a causal factor, to confound the outcome estimate.

Answer: Thank you for the opportunity to clarify this point. We conducted a review of the literature and generally speaking, there is no strong evidence to support that pregnancy use of the medications mentioned by the reviewer are causal for both neurodevelopmental disorders and ASM use. As the reviewer might anticipate, it is challenging to provide references to support the

absence of a causal effect of other medications on antiseizure treatments—largely because no study would likely examine this, given the lack of a theoretical basis for expecting such an effect.

However, there is better evidence that indications for those medications (e.g., bipolar disorder for lithium, ADHD for stimulant medications) are linked to offspring neurodevelopmental disorders. Where possible, we have accounted for these conditions in our models by: 1) including them explicitly as covariates (e.g., bipolar disorder); 2) adjusting indirectly for healthcare visits; and 3) using sibling analyses to further control for familial confounding.

Finally, as we already acknowledged in the limitations section (see line 385 to 387), our study is observational and, as such, may be subject to residual confounding. Our findings, like those of any study on this topic, should be interpreted with this consideration in mind.

6. In response to comment 12, please consider presenting the results in supplementary results.

Answer: We have now included the results without adjustment for antiemetics in the supplementary materials and have revised the main text accordingly. Please see the new supplementary Table S14 below:

Table S14 – Sensitivity analysis: Comparison of primary analysis models including and excluding vomiting or antiemetics as a covariate

Outcome	ASM	CPRD (UK)		DOHaD (Sweden)		Pooled	
		Primary	Sensitivity *	Primary	Sensitivity *	Primary	Sensitivity *
Autism	No ASM	1 (Reference)	1 (Reference)	1 (Reference)	1 (Reference)	1 (Reference)	1 (Reference)
	Carbamazepine	1.55 (0.94-2.53)	1.53 (0.94-2.51)	1.21 (1.01-1.45)	1.21 (1.01-1.45)	1.25 (1.05-1.48)	1.24 (1.05-1.48)
	Gabapentin	1.43 (0.88-2.32)	1.47 (0.91-2.39)	0.97 (0.64-1.50)	0.98 (0.64-1.50)	1.15 (0.84-1.59)	1.17 (0.85-1.61)
	Lamotrigine	1.10 (0.67-1.78)	1.09 (0.67-1.78)	0.83 (0.69-0.99)	0.83 (0.69-0.99)	0.86 (0.72-1.01)	0.86 (0.72-1.01)
	Levetiracetam	0.90 (0.22-3.67)	0.87 (0.21-3.58)	0.80 (0.40-1.62)	0.80 (0.40-1.62)	0.82 (0.44-1.54)	0.82 (0.44-1.53)
	Phenytoin	1.19 (0.18-8.05)	1.19 (0.18-8.01)	0.97 (0.52-1.82)	0.98 (0.52-1.82)	0.99 (0.55-1.80)	0.99 (0.55-1.80)
	Pregabalin	0.79 (0.37-1.67)	0.81 (0.38-1.71)	0.70 (0.53-0.93)	0.70 (0.53-0.94)	0.71 (0.54-0.93)	0.71 (0.55-0.94)
	Topiramate	0.31 (0.04-2.21)	0.31 (0.04-2.26)	1.26 (0.73-2.18)	1.28 (0.74-2.21)	1.14 (0.68-1.93)	1.16 (0.68-1.96)
	Valproate	1.51 (0.87-2.63)	1.50 (0.86-2.61)	1.82 (1.49-2.21)	1.82 (1.49-2.21)	1.78 (1.48-2.14)	1.78 (1.48-2.14)
	Other	1.63 (0.52-5.12)	1.59 (0.50-4.99)	1.28 (0.89-1.83)	1.28 (0.89-1.83)	1.30 (0.93-1.84)	1.30 (0.92-1.84)
Polytherapy	1.55 (0.94-2.55)	1.55 (0.94-2.56)	1.51 (1.19-1.91)	1.51 (1.19-1.92)	1.51 (1.22-1.88)	1.52 (1.22-1.88)	
Intellectual disability	No ASM	1 (Reference)	1 (Reference)	1 (Reference)	1 (Reference)	1 (Reference)	1 (Reference)
	Carbamazepine	1.67 (0.58-4.85)	1.67 (0.58-4.84)	1.28 (0.98-1.68)	1.28 (0.98-1.68)	1.30 (1.01-1.69)	1.30 (1.01-1.69)
	Gabapentin	-	-	1.02 (0.49-2.15)	1.02 (0.49-2.15)	1.02 (0.49-2.15)	1.02 (0.49-2.15)
	Lamotrigine	1.57 (0.57-4.38)	1.56 (0.56-4.35)	1.09 (0.83-1.44)	1.09 (0.83-1.44)	1.12 (0.86-1.46)	1.12 (0.86-1.46)
	Levetiracetam	-	-	0.66 (0.21-2.05)	0.66 (0.21-2.05)	0.66 (0.21-2.05)	0.66 (0.21-2.05)
	Phenytoin	-	-	0.67 (0.22-2.10)	0.67 (0.22-2.10)	0.67 (0.22-2.10)	0.67 (0.22-2.10)
	Pregabalin	-	-	0.86 (0.52-1.40)	0.86 (0.52-1.40)	0.86 (0.52-1.40)	0.86 (0.52-1.40)
	Topiramate	3.12 (0.43-22.82)	3.18 (0.43-23.49)	2.40 (1.14-5.06)	2.41 (1.14-5.07)	2.48 (1.23-4.98)	2.49 (1.24-5.00)
	Valproate	1.04 (0.25-4.34)	1.03 (0.25-4.31)	2.64 (2.02-3.44)	2.64 (2.02-3.44)	2.56 (1.97-3.32)	2.56 (1.97-3.32)
	Other	-	-	0.90 (0.45-1.81)	0.90 (0.45-1.81)	0.90 (0.45-1.81)	0.90 (0.45-1.81)
Polytherapy	1.95 (0.70-5.43)	1.96 (0.70-5.47)	2.03 (1.47-2.80)	2.03 (1.48-2.80)	2.02 (1.49-2.75)	2.03 (1.49-2.75)	
Attention Deficit Hyperactivity Disorder	No ASM	1 (Reference)	1 (Reference)	1 (Reference)	1 (Reference)	1 (Reference)	1 (Reference)
	Carbamazepine	1.51 (0.87-2.61)	1.50 (0.86-2.60)	1.05 (0.92-1.19)	1.05 (0.92-1.19)	1.07 (0.94-1.21)	1.07 (0.94-1.21)
	Gabapentin	1.77 (0.92-3.43)	1.85 (0.95-3.58)	1.13 (0.82-1.54)	1.13 (0.83-1.55)	1.22 (0.92-1.63)	1.24 (0.93-1.64)
	Lamotrigine	1.64 (0.93-2.89)	1.63 (0.92-2.87)	0.86 (0.75-0.98)	0.86 (0.75-0.98)	0.89 (0.78-1.01)	0.89 (0.78-1.01)
	Levetiracetam	1.48 (0.28-7.83)	1.43 (0.27-7.47)	1.02 (0.60-1.72)	1.01 (0.60-1.72)	1.05 (0.64-1.74)	1.05 (0.63-1.73)
	Phenytoin	2.15 (0.30-15.39)	2.13 (0.30-15.27)	1.13 (0.77-1.67)	1.13 (0.77-1.67)	1.16 (0.79-1.70)	1.16 (0.79-1.70)
	Pregabalin	0.92 (0.29-2.93)	0.95 (0.30-3.02)	0.67 (0.54-0.82)	0.67 (0.54-0.82)	0.67 (0.55-0.83)	0.68 (0.55-0.83)
	Topiramate	0.71 (0.10-5.32)	0.72 (0.10-5.40)	0.80 (0.48-1.33)	0.81 (0.49-1.34)	0.80 (0.49-1.30)	0.80 (0.49-1.31)
	Valproate	1.48 (0.71-3.07)	1.46 (0.70-3.03)	1.18 (1.01-1.39)	1.19 (1.01-1.39)	1.20 (1.02-1.40)	1.20 (1.02-1.40)
	Other	1.05 (0.14-7.77)	1.01 (0.14-7.52)	0.94 (0.71-1.26)	0.94 (0.71-1.26)	0.95 (0.71-1.25)	0.95 (0.71-1.25)
Polytherapy	1.13 (0.52-2.45)	1.12 (0.52-2.44)	1.12 (0.92-1.37)	1.12 (0.92-1.37)	1.12 (0.92-1.36)	1.12 (0.92-1.36)	

* Sensitivity excluding vomiting or antiemetics as a covariate

7. In response to comment 17, the authors revised the final paragraph under strengths and limitations, “Although our combined efforts across the UK and Sweden have allowed us to amass one of the largest and longest follow-ups of children diagnosed with neurodevelopmental conditions following ASM exposure in pregnancy, we emphasize that further data on ASM safety are warranted. This need becomes more pertinent as the landscape of ASM continues to evolve, given changing indication prevalences, costs, and known teratogenicity profiles. To enable such large-scale analysis and enhance generalizability, it will remain crucial to involve data from multiple countries across the globe.”

The highlighted part is not true, compared to the quoted Nordic study, which has a larger sample size and longer follow-up time.

Answer: Thank you for allowing us to clarify. As we mentioned in response to Reviewer Comment 1, while the Nordic study did have a large sample size, it had a shorter follow-up period and fewer exposed cases. We believe our use of the phrase "follow-ups of children diagnosed with neurodevelopmental conditions following ASM exposure" accurately reflects this distinction.

8. In response to comment 20, if the Swedish registry uses a broader definition for somatic conditions, why was this definition not applied to the UK database? Additionally, it's important to note that the Swedish database contributed a substantial weight to the majority of the analyses. This discrepancy may warrant mentioning in the manuscript.

Answer: Thank you for allowing us to expand on this. After agreeing on what we deemed to be important confounders, we attempted to use ‘the best possible approach’ in each country. That is, we felt that it was better to try to capture the confounders to the best of each country's ability rather than compromise a site with better resolution to the level of the one with poorer resolution. The upside of this strategy is that each country can provide as valid estimates as possible. In response to your comment 2 above, we have tried to make the approach clearer in the manuscript.

As we have already outlined in the manuscript, the downside of this strategy is that it will result in differences in precise covariate definitions. Despite these, however, we find that the estimates are largely consistent across the countries, lending some validity to our findings.

We agree that Sweden has a substantial weight in the majority of the analyses, but that’s also why we have made extensive efforts to report country-specific estimates (e.g., See Table S1, S4, S6, S8, etc.), and provide a detailed analysis of heterogeneity across the countries (e.g., See Table S7); which we find to be minimal.

9. It is recommended to add footnote, explaining all the abbreviations used in the all tables, including ones in the supplementary results.

Answer: Thank you for bringing this to our attention. We have reviewed all tables and ensured that all abbreviations are clarified in the footnotes.

Reviewer #3 (Remarks to the Author):

Many thanks to the authors for their thorough review and response to all of the comments submitted in the first review stage. The paper is very detailed in its methods and presentation of results and reads clearly and well. The revisions have mostly addressed the areas that concerned me but the following remain as points that could be updated or further clarified in the text:

Answer: We are pleased that the reviewers appreciated our revisions, and we are grateful for their attention to detail. We have carefully addressed and revised their remaining comments, and we agree that the rigor of our study has been strengthened thanks to the reviewers' valuable input.

Intellectual disability: thank you for including the definition of this diagnosis. As the definition of this has changed over time to not just include IQ scores, please would you also include the code lists (or a link to a repository holding these) to aid others in comparing or undertaking to report similar work.

Answer: Thank you for this suggestion. We have included all code lists within a published GitHub repository (<https://github.com/pmadleydowd/PREPAR-E-ASM-Neurodevelopment>, with DOI: <https://doi.org/10.5281/zenodo.13820418>). This link and a reference to the DOI are now contained in the code availability section after the methods.

Polytherapy: I recognise that many combinations of ASMs may be present in the data; given the findings for valproate, carbamazepine and topiramate, please could you indicate if any of the results for polytherapy include any of these three ASMs? This is very important to understand in terms of the potential influence that these three ASMs could be having on the polytherapy results.

Answer: Thank you for the opportunity to expand on the polytherapy results. We agree that further clarification enhances the interpretation. In the below table, we provide the counts for individual antiseizure medications used in the polytherapy group, which indicate that the observed excess risks are likely attributable to the higher usage rates of valproate and carbamazepine in this group.

We have now included the table with these counts in the appendix, and we reference it concisely in the results section. Additionally, we have expanded our discussion to briefly acknowledge this potential explanation (second paragraph of the Comparison with previous literature section in the Discussion):

“Specific polytherapy combinations, including levetiracetam with carbamazepine and lamotrigine with topiramate, have been suggested to be associated with neurodevelopmental disorders. Our results suggest an increased risk of autism and intellectual disability in the children of mothers exposed to polytherapy during pregnancy, though we did not explore specific combinations. **As valproate and carbamazepine were more commonly used in the polytherapy-exposed group, this may account for the observed excess risks (see Supplementary Results Table S15).** It is worth caveating that such findings may be particularly susceptible to confounding by severity of indication as mothers taking more than one ASM may be more likely to suffer from more severe epilepsy or other indications than mothers taking monotherapy.”

Supplementary Table S15: Counts of antiseizure medications included in the polytherapy category

ASM included in polytherapy	CPRD (UK), N (%) with polytherapy = 573 (100.00%)	DOHaD (Sweden), N (%) with polytherapy = 1167 (100.00%)
Carbamazepine	204 (35.6%)	353 (30.2%)
Gabapentin	65 (11.3%)	92 (7.9%)
Lamotrigine	268 (46.8%)	794 (68.0%)
Levetiracetam	225 (39.3%)	449 (38.5%)
Phenytoin	27 (4.7%)	36 (3.1%)
Pregabalin	49 (8.6%)	187 (16.0%)
Topiramate	57 (10.0%)	157 (13.5%)
Valproate	145 (25.3%)	347 (29.7%)
Other	205 (35.8%)	0 (0.0%)

Note that rows within a column are not mutually exclusive and so the sum of percentages will be greater than 100%

Pg14, line 410: the data from the MHRA review is based on field studies rather than database studies. This highlights the difficulty that database studies can have in capturing all outcomes of this type - sometimes because people are lost to follow up or diagnoses take a long time to be made. You have noted this already in a previous paragraph therefore an update to indicate the differences in the types of studies and results found would help to explain this disparity.

Answer: Thank you for allowing us to expand on this matter. The text that MHRA (reference 45) published read as follows:

“Valproate has a high teratogenic potential. Exposure of an unborn child to valproate in utero is associated with a high risk of congenital malformations (11%) and neurodevelop-

mental disorders (30–40%), which may lead to permanent disability. The available evidence does not support a specific at-risk gestational period and the possibility of a risk of valproate throughout pregnancy cannot be excluded.”

Considering your comment, we went back and reviewed the MHRA information. We find that the “30-40%” statements originate from the MHRA document “Valproate Pregnancy Prevention Programme” (<https://www.gov.uk/drug-safety-update/valproate-medicines-epilim-depakote-contraindicated-in-women-and-girls-of-childbearing-potential-unless-conditions-of-pregnancy-prevention-programme-are-met#fn:1>). In this document/report, they provide the following references:

- 1) [https://doi.org/10.1016/S1474-4422\(12\)70323-X](https://doi.org/10.1016/S1474-4422(12)70323-X)
- 2) <https://doi.org/10.1001/jama.2013.2270>
- 3) <http://www.doi.org/10.1111/epi.12226>
- 4) <https://doi.org/10.1016/j.yebeh.2013.08.001>
- 5) <https://doi.org/10.1111/j.1528-1167.2010.02668.x>
- 6) <http://dx.doi.org/10.1136/adc.2009.176990>
- 7) <http://www.doi.org/10.1056/NEJMoa0803531>
- 8) <https://doi.org/10.1016/j.yebeh.2008.01.010>
- 9) <http://dx.doi.org/10.1002/14651858.CD010236.pub2>

To see how MHRA arrives at 30-40% we reviewed all these studies. Briefly, these studies consist of cohorts of small sizes or large cohorts with linkage to registry information like that used in our study. In many cases, other measures of neurodevelopment, such as cognitive ability, are used. We are unable to identify any support for the 30-40% statistic cited by the MHRA, or the methods used to achieve it. We speculate that the statement is a confusion about relative risks (e.g., RR 1.30-1.40) and absolute risks.

Nevertheless, we have revised the text on Page 14, line 410 to read as follows:

“The MHRA has stated that the risk of NDD is 30-40% following exposure to valproate [45]. We find substantially lower absolute risks in the UK and Sweden and corroborate results from the Nordic based SCAN-AED study [14, 15]. The differences in estimates with those presented by the MHRA may be explained by our use of electronic health records and registry data with increased sample sizes and lengths of follow up as well as by different measures used to assess neurodevelopment. The difference also highlights the need for better quality evidence and communication of risk in this area.”